# Mediodorsal thalamic nucleus mediates resistance to ethanol through Ca$_v$3.1 T-type Ca$^{2+}$ regulation of neural activity

Charles-francois V Latchoumane[1], Joon-Hyuk Lee[1], Seong-Wook Kim[1], Jinhyun Kim[2], Hee-Sup Shin[1]*

[1]Institute for Basic Science, Center for Cognition and Sociality, Daejeon, Republic of Korea; [2]Korea Institute of Science and Technology, Center for Functional Connectomics, Seoul, Republic of Korea

*For correspondence:
shin@ibs.re.kr

Competing interest: The authors declare that no competing interests exist.

## eLife Assessment

This **valuable** study investigates the relationship between neuronal dynamics in the thalamus and brain state modulation. The claims that a specific channel is a critical player in the regulation of brain-states and ethanol-resistance in mice are supported by **convincing** evidence. The work will be of interest to systems neuroscientists interested in brain dynamics and behavioural states.

**Abstract** Thalamocortical activity is known to orchestrate sensory gating and consciousness switching. The precise thalamic regions involved, or the firing patterns related to the unconsciousness, remain unclear. Interestingly, the highly -expressed thalamic T-type calcium currents have been considered as a candidate for the ionic mechanism for the generation of thalamocortically driven change in conscious state. Here, we tested the hypothesis that Ca$_v$3.1 T-type channels in the mediodorsal thalamic nucleus (MD) might control neuronal firing during unconsciousness using Ca$_v$3.1 T-type channel knockout (KO) and knockdown (KD) mice under natural sleep and ethanol-induced unconsciousness. During natural sleep, the MD neurons in KO mice showed general characteristics of sustained firing across sleep stages. We found that KO and MD-specific KD mice showed enhanced resistance to ethanol. During the ethanol-induced unconscious state, wild-type (WT) MD neurons showed a significant reduction in neuronal firing from baseline with increased burst firing, whereas Ca$_v$3.1 KO neurons showed well-sustained neural firing, within the level of wakefulness, and no burst firing. Further, 20 Hz optogenetic and electrical activation of MD neurons mimicked the ethanol resistance behavior in WT mice. These results suggest that maintaining MD neural firing at a wakeful level is sufficient to induce resistance to ethanol-induced hypnosis in WT mice. This work has important implications for the design of treatments for consciousness disorders using thalamic stimulation of deeper nuclei, including the targeting of the mediodorsal thalamic nucleus.

## Introduction

Drug-induced unconsciousness can be achieved using numerous types of anesthetics with varying modes of action (*Alkire and Miller, 2005*; *Mashour, 2014*). Ethanol, one of the most frequently abused drugs in human society, can induce sleep-like loss of consciousness at high doses (*Morozova et al., 2014*). While possible neuropharmacological and neural correlates of ethanol sedation have been proposed using in vitro and in vivo methods (*Blomeley et al., 2011*; *Givens and Breese, 1990*; *Koob et al., 2013*; *White et al., 1990*), recent studies have highlighted the slowing of thalamocortical-driven rhythms as a potent marker of unconsciousness (*Baker et al., 2014*; *Schiff, 2008*). However,

the region and the mechanism linked to thalamic modulation during ethanol-induced unconsciousness remain poorly understood.

Physiological correlates of thalamocortical rhythmic activities and consciousness state of the brain have been studied extensively (*Llinás et al., 1998*; *Alkire et al., 2008*; *Steriade et al., 1993*; *Kim et al., 2001*). T-type calcium channels are known generators of thalamocortical rhythms, through the modulation of cell excitability and rebound burst firing (*Simms and Zamponi, 2014*; *Crunelli et al., 2018*). During sleep, the transition from wakefulness to unconsciousness is associated with membrane hyperpolarization of thalamic neurons (*Steriade et al., 1993*; *Royer et al., 2012*). Similarly, it has been shown that in ethanol sedation, as in natural sleep or absence seizure, the loss of consciousness is characterized by a switch from tonic to burst firing in thalamic neurons, which involves GABAergic inhibition-driven de-inactivation of $Ca_v3.1$ (*Cacna1g*) T-type channels resulting in slow oscillatory response of the thalamocortical network (*White et al., 1990*; *Jia et al., 2008*; *Jia et al., 2007*; *Iyer et al., 2011*). Acute intoxication at high doses of ethanol (*Newton et al., 2008*; *Shin et al., 2005*) induces both slow oscillations in the delta-theta frequency range and a loss of righting reflex (LORR), a classical proxy to assess the loss of consciousness. It has been shown that mice lacking global or thalamic $Ca_v3.1$ showed altered slow oscillations and sleep architecture (*Anderson et al., 2005*; *Lee et al., 2004*); delayed sleep induction under several anesthetics (i.e. isoflurane, halothane, sevoflurane, and pentobarbital) (*Petrenko et al., 2007*); and increased resistance to drug-induced absence seizures (*Kim et al., 2001*). Notably, the absence or blockade of $Ca_v3.1$ resulted in an increased preference for ethanol consumption and novelty-seeking behavior (*Newton et al., 2008*; *Shin et al., 2005*). In the current studies, we investigate the role of Ca3.1-mediated T-currents in brain state modulation during ethanol-induced sleep.

The thalamus is one of the major regions expressing $Ca_v3.1$ T-type calcium channels (*Talley et al., 1999*) and holds a central role in information transmission and integration (*Schiff et al., 2014*). In vitro and in vivo studies using genetically modified mice have revealed that $Ca_v3.1$ T-type channels play a key role in the genesis of thalamocortical rhythms, such as 3 Hz spike-and-wave discharges, a signature of absence seizures (*Kim et al., 2001*; *Song et al., 2004*) and delta waves (*Royer et al., 2012*; *Lee et al., 2013*; *McCormick and Pape, 1990*). Previous investigations on thalamic control of consciousness revealed that nuclei within the dorsal medial thalamus (dMT) hold an important modulatory function in the interaction of attention and arousal (*Schiff, 2008*; *Saalmann, 2014*). Particularly, the centromedian (CM) thalamic nucleus, and not the ventrobasal nucleus (VB), showed rapid shifts in local field potential (LFP) preceding brain state transitions such as non-rapid eye movement (NREM) and propofol-induced anesthesia (*Baker et al., 2014*). The paraventricular thalamic nucleus (PVN) showed critical involvement in wake/sleep cycle regulation (*He et al., 2015*). The centrolateral (CL) thalamic nucleus has been implicated in the modulation of arousal, behavior arrest (*Giber et al., 2015*), and improvement of level of consciousness during seizures (*Gummadavelli et al., 2015*). Notably, the direct electrical stimulation of the intralaminar nuclei (ILN) and, in particular, CL, promoted hallmarks of arousal and awakening in primates under propofol and ketamine propofol anesthesia. The MD, a subnucleus of dMT, on the other hand, has only recently been implicated in disorders of consciousness (*He et al., 2015*) and ketamine/ethanol-induced loss of consciousness (*Choi et al., 2015*) through the alteration of thalamocortical functional connectivity. In anesthetized primates, the stimulation of ILN and MD increased arousal and wakefulness score (*Bastos et al., 2021*). However, several key questions remain to be answered: (1) Is there a specific role for MD $Ca_v3.1$ T-type calcium channels in the control of ethanol-induced loss of consciousness? (2) Does $Ca_v3.1$ T-type calcium channel-driven neuronal firing pattern have any role in the control of consciousness?

In this study, we identified that knockout (KO) and MD-specific silencing of $Ca_v3.1$ T-type calcium channels results in increased ethanol resistance in mice. Using single-unit recordings, we compared MD activity of wild-type (WT) and KO mice while the mice transitioned from conscious to unconscious state and found that the KO mice showed more sustained MD activity, whereas the WT mice showed clearly reduced MD activity. Furthermore, and consistently with their resistant phenotype, KO mice showed sustained MD firing, well within the wakefulness level, under ethanol consumption. Finally, we demonstrate that both the optogenetic and electrical stimulations in MD, mimicking the sustained firing pattern of KO mice, were sufficient to induce the increased ethanol resistance in WT mice. These results reveal a causal control of brain state by MD during ethanol-induced unconsciousness and with an underlying neural mechanism governed by $Ca_v3.1$ T-type calcium channels.

## Results

To understand the role of T-type $Ca^{2+}$ channels in modulating the consciousness level, we compared the ethanol resistance between WT and $Ca_v$3.1 KO (*Cacna1g−/−*) littermates. We used the forced walking task (FWT; *Figure 1A*), an analog to the LORR assay, which enables a continuous and high-temporal resolution assessment of the loss of movement (*Hwang et al., 2010*) (LOM). Moreover, the FWT objectively measures the latency to and duration of the first LOM, but also the total time spent in LOM using automatized analysis of video confirmed by electromyograms (EMGs) or accelerometer recordings (*Figure 1B* and *Figure 1—figure supplement 1*; see Methods). The continuously running treadmill (6 cm/s) ensures a normalized behavior within and between animals before injection (i.e. baseline forced walking) and allows for reduced intervention from experimenters for the monitoring of both electrophysiological (*Figure 1—figure supplement 1*, upper panel) and analyzed behavioral (*Figure 1—figure supplement 1*, lower panel).

### The lacking $Ca_v$3.1 increases ethanol resistance in mice

Mice lacking $Ca_v$3.1 exhibited delayed anesthetic induction (*Kim et al., 2001*; *Petrenko et al., 2007*) and impairment in maintenance of low conscious level (*Anderson et al., 2005*; *Lee et al., 2004*), as well as increased ethanol preference (*Shin et al., 2005*; *Choi et al., 2015*). We tested the sensitivity of $Ca_v$3.1 null mutant mice for various acute hypnotic doses of ethanol.

We confirmed that the lack of $Ca_v$3.1 resulted in a more delayed and fragmented LOM (*Figure 1C1 and C2*), and a reduction in the total time spent in LOM compared to $Ca_v$3.1 WT mice (*Figure 1C3*). We observed that $Ca_v$3.1 null mutant mice showed increased latency to and decreased duration of the first episode of loss of movement (fLOM) for ethanol injection doses of 2.0, 3.0, and 4.0 g/kg (*Figure 1—figure supplement 2*). The total time spent in LOM during 1 hr of recording was also significantly reduced (*Figure 1—figure supplement 2A3 and B3*) compared with WT mice. Two-way analysis of variance (ANOVA) showed a significant effect for the main factor genotype and dose for the latency to and duration of fLOM and total time in LOM (*Figure 1—source data 1*), indicating a dose dependency in both wild and mutant mice. In particular, an intraperitoneal (i.p.) injection of 3.0 g/kg induced a significant difference between wild and mutant mice in latency to (t(16) = −4.1965, p=0.002; Student's t-test) and duration of fLOM (t(16) = 2.3908, p=0.0294; Student's t-test), and total time spent in LOM (t(16) = 3.9065, p=0.0012; Student's t-test).

These results indicate that ethanol induces a dose-dependent sedative effect on mice, and $Ca_v$3.1 mutant mice had an increased resistance to ethanol sedation compared to WT mice.

### $Ca_v$3.1 silencing in the MD, but not VB, increased ethanol resistance in mice

A great majority of $Ca_v$3.1 expression is found in the thalamic region (*Talley et al., 1999*) and was shown to specifically correlate with the modulation of thalamocortical-related rhythms and stability of sleep level (*Anderson et al., 2005*; *Lee et al., 2004*). To identify a possible role of the thalamic region in ethanol resistance, we knocked down the expression of $Ca_v$3.1 in the MD and a VB region, two regions possibly involved in a thalamic control of consciousness (*He et al., 2015*; *Choi et al., 2015*; *Cheong et al., 2009*), using a lentivirus (LV)-mediated short hairpin (shRNA) delivery.

We found that compared to shControl-injected mice, $Ca_v$3.1 knockdown (KD) of MD resulted in an increased latency to (*Figure 2A1*; t(27) = −3.0045, p=0.0057; Student's t-test) and duration of (*Figure 2A2*; t(27) = 2.1448, p=0.0411; Student's t-test) fLOM, and total time spent in LOM (*Figure 2A3*; t(27) = 2.6641, p=0.0128, two-tailed test) for 3.0 g/kg i.p. injection of ethanol. However, we found that compared to shControl-injected mice, $Ca_v$3.1 KD of VB did not change the latency to (t(8) = −1.0093, p=0.3423, two-tailed test), duration of (t(8) = −0.0983, p=0.9241, two-tailed test) fLOM and total time spent in LOM (t(8) = −0.6317, p=0.5452, two-tailed test) for the same 3.0 g/kg i.p. injection of ethanol (*Figure 2—figure supplement 1*). Representative traces of mice activity showed that mice with $Ca_v$3.1 KD in MD (*Figure 2—figure supplement 2A*) had a more delayed and fragmented early period of LOM compared to MD LV-shControl- and VB-injected mice, as in mutant mice.

We then characterized the change in $Ca_v$3.1 expression following the shControl and shCa_v3.1 KD injections in three test regions: MD (left and right), CM, and CL (left and right side), and a negative control region SMT (submedial thalamic nuclei, left and right side). The average intensity was

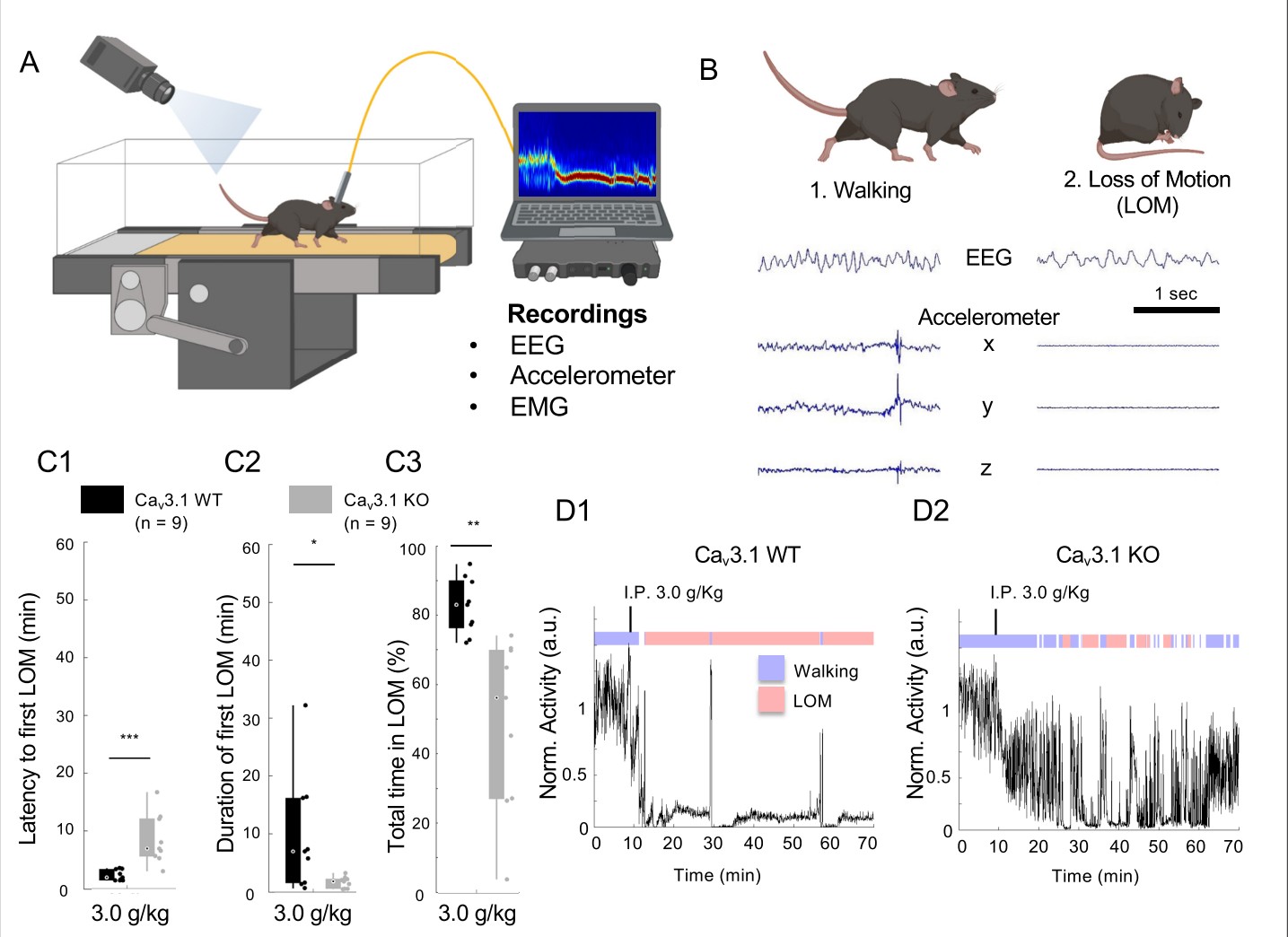

**Figure 1.** Mice lacking Ca$_v$3.1 showed increased ethanol resistance on the forced walking task (FWT). (**A**) The schematic of the FWT setup. Mice are habituated and trained on a constantly moving treadmill (6 cm/s). Following a baseline walking recording (~10 min), the mouse is carefully picked up and injected with ethanol (i.p.). Once placed back on the treadmill, the loss of consciousness is evaluated using normalized moving index using either video analysis (differential pixel motion), on-head accelerometer-based motion, or neck electromyograms over a period of 60 min. (**B**) Representative EEG (parietal) and three-axis accelerometer (Acc) traces for walking and loss of movement (LOM) in a Ca$_v$3.1 wild-type (WT) mouse. (**C**) Quantification for the latency to first LOM (fLOM; i.e. delay between i.p. injection and LOM; **C1**), the duration of the fLOM (**C2**), and the total time spent in LOM state (**C3**) over a recording duration of 60 min following 3.0 g/kg i.p. injection of ethanol in Ca$_v$3.1 WT and Ca$_v$3.1 knockout (KO) mice; data is represented as boxplot with individual mice as scatter plot. * is for $p<0.05$, ** is for $p<0.01$, and *** is for $p<0.001$. (**D**) Representative normalized motor activity over time for Ca$_v$3.1 WT (**D1**) and Ca$_v$3.1 KO (**D2**) mice post i.p. injection of 3.0 g/kg. Blue and red boxes above the graph indicate the state interpretation for walking and LOM, respectively.

The online version of this article includes the following source data and figure supplement(s) for figure 1:

**Source data 1.** Two-way analysis of variance (ANOVA) for latency to and duration of first loss of movement (fLOM), and total time spent in LOM; main factor gene, two levels (Ca$_v$3.1 wild-type [WT] and Ca$_v$3.1 knockout [KO]); main factor dose, three levels (2.0 g/kg, 3.0 g/kg, and 4.0 g/kg).

**Figure supplement 1.** The forced walking task (FWT) is an uninterrupted assessment of the mouse sedative state equivalent to the loss of righting reflex (LORR).

**Figure supplement 2.** Ca$_v$3.1 mutant mice show a dose-dependent ethanol resistance.

obtained from two coronal brain slices for each mouse used in the experiment (see Methods sections, Ca$_v$3.1 Intensity quantification). Our results show that the targeting of the KD was very specific to the bilateral MD ($p<0.001$; *Figure 2D*). We noted that the CM ($p<0.05$) and a marginal unilateral KD of the CL were also observed ($p<0.01$). Notably, we tested the correlation between the level of KD in MD and the total time in LOM and observed a significant association (*Figure 2D* inset; R=0.599,

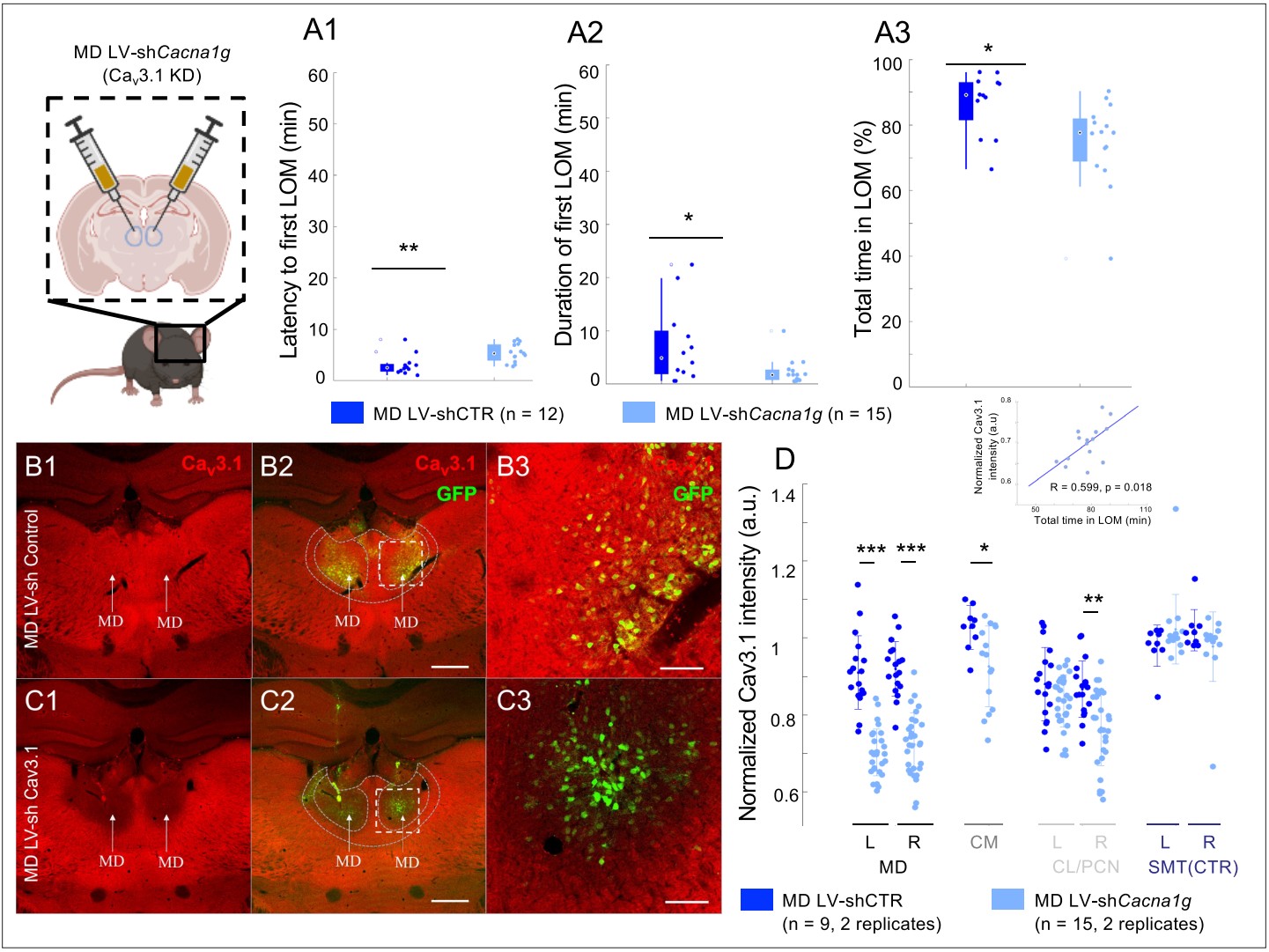

**Figure 2.** Ca$_v$3.1 knockdown in mediodorsal thalamic nucleus (MD) increased ethanol resistance in mice. (**A**) Quantification for the latency to first loss of movement (fLOM; **A1**), the duration of the first LOM (**A2**), and the total time spent in LOM state (**A3**) over a recording duration of 60 min post intraperitoneal (i.p.) injection of ethanol (3.0 g/kg) in lentivirus (LV)-shControl and sh*Cacna1g* (Ca$_v$3.1 knockdown) mice for MD; data is represented as a boxplot with individual mice shown as a scatter plot. * and ** indicate p<0.05 and p<0.01, respectively. (**B**) Representative brain coronal section stained using Ca$_v$3.1 antibody (**B1**), showing the endogenous thalamic expression of Ca$_v$3.1 in the MD of LV-shControl-injected mice; Ca$_v$3.1 and GFP merging (**B2**) and higher magnification of the white dashed square in B2 (**B3**). Scale bars in (**B2**) and (**B3**) indicate 500 μm and 100 μm, respectively. (**C**) Representative brain coronal section showing the reduced Cav3.1 expression in the MD of LV-sh*Cacna1g* (Ca$_v$3.1 knockdown) injected mice (**C1**); Ca$_v$3.1 and GFP positive merging (**C2**) and higher magnification of the white dashed square in C2 (**C3**). Scale bars in (**C2**) and (**C3**) indicate 500 μm and 100 μm, respectively. (**D**) Normalized Ca$_v$3.1 intensity estimated for the nuclei MD, CM (centromedian), CL/PCN (centrolateral/paracentral) and SMT (submedial thalamic nucleus). The quantification was performed as intensity per area for two replicates per side per mouse. *, **, and *** indicate p<0.05, p<0.01, and p<0.001, respectively (two-sample t-test). The data is shown as a scatter plot for all values and superposed with the mean and standard deviation error bars. Inset: We noted a positive correlation between the total LOM duration and the Ca$_v$3.1 intensity in MD (R=0.599, p=0.018).

The online version of this article includes the following figure supplement(s) for figure 2:

**Figure supplement 1.** Mice with ventrobasal nucleus (VB) Ca$_v$3.1 knockdown (KD) did not show increased ethanol resistance.

**Figure supplement 2.** Representative activity of mice with Ca$_v$3.1 knockdown (KD) in mediodorsal thalamic nucleus (MD) and ventrobasal nucleus (VB).

---

p=0.018). This result highlights that the Ca$_v$3.1 KD was specific to MD and with an intensity associated with ethanol-induced LOM.

During the open-field test, Ca$_v$3.1 null mutant mice showed significantly increased locomotor activity compared to WT mice as shown by total distance moved (*Figure 2—figure supplement 2C*; ANOVA: GROUP F(3) = 8.45, p=0.0004; Ca$_v$3.1 WT vs Ca$_v$3.1 KO: p=0.0001). The mice with

MD-specific Ca$_v$3.1 KD, however, did not show any significant difference in total distance moved compared to shControl-injected control mice (MD LV-shControl vs MD LV-sh*Cacna1g* [Ca$_v$3.1 KD]: p=0.868; Holm-Sidak correction), indicating that the ethanol resistance in MD Ca$_v$3.1 KD mice was not attributed to hyperlocomotion observed in Ca$_v$3.1 KO mice.

## Lack of Ca3.1 in MD neurons removes thalamic burst in NREM sleep

Thalamic neurons are known to follow a state-dependent activity (*Poulet et al., 2012*); however, the nature of this state-dependent activity has not been studied for the MD. In order to understand the relationship between MD neuron firing and level of consciousness, we investigated the association between the neural activity in MD and brain states at different levels of consciousness (*Figure 3*, *Figure 3—figure supplement 1*).

We observed two major populations of neuronal spike waveform present in MD single-unit recordings of Ca$_v$3.1 WT mice, also described in previous works (*Schiff and Reyes, 2012*; *Destexhe, 2009*): (1) a majority of regular spiking (RS) cells characterized by wide spike waveform (36/39 neurons, 92.3%), i.e., spike-to-valley width >250 µs (*Figure 3B*) and high bursting propensity; (2) a minority of narrow spiking (NS; also known as fast spiking) cells showing short spike-to-valley width, i.e., <250 µs, lower bursting characteristics (*Figure 3*, *Figure 3—figure supplement 2A3 and A4*) and fast-paced tonic firing (10–50 Hz; data not shown). The RS and NS neurons were found in MD of WT and mutant mice; however, MD RS mutant neurons showed an absence of short inter-spike intervals (ISI, i.e. indicative of the absence of burst) in auto-cross-correlogram (*Figure 3—figure supplement 2*) and a clear reduction in total bursting represented as bursting index (*Figure 3B*; ratio of spikes count <10 ms and >50 ms based on autocross-correlogram). Since RS cells have the profile of the major population of MD, i.e., excitatory neurons, we focused on the analysis of RS neurons mainly in the remainder of this study.

During the deep sleep state NREM, thalamic neural firing is known to switch from tonic to burst firing (*Llinás and Steriade, 2006*). We found that a lack of Ca$_v$3.1 T-type calcium channels resulted in a near absence of burst (see Methods for definition) in mutant mice (*Figure 3C1 and C2*; 4/44 bursting neurons; Z(77) = 7.20, p<0.0001, rank-sum test) compared to WT mice (34/34 bursting neurons; 5.76±5.51 burst events/min).

## Lack of Ca3.1 reduces neuronal activity across all brain states in MD

In addition, we observed that the mutant mice showed a significantly lower total firing rate (main factor group: F(1,186) = 16.5, p=0.0001; interaction group × brain state: F(3,186) = 4.72, p=0.0034) and a reduced variability (p<0.0001 for all brain states; Levene's test) in most brain states compared to the WT mice, indicating that the lack of Ca$_v$3.1 T-type channels results in an overall reduction in neural activity in RS neurons.

RS neurons of MD in Ca$_v$3.1 WT and mutant mice showed a significant change in overall firing across walking, waking (home cage), NREM, and REM sleep states as shown by a repeated-measures ANOVA (*Figure 3D*; main factor brain state: F(3,186) = 104.96, p<0.0001). In addition, Ca$_v$3.1 WT (R=–0.534, p=1.6e-10, Spearman's rank correlation) and mutant (R=–0.689, p=6.8e-20, Spearman's rank correlation) showed a significant negative correlation between neural firing and brain state. Assuming an ordering from higher to lower state of consciousness, these results indicate that MD firing is associated with the level of consciousness independently from the Ca$_v$3.1 T-type channels in WT and mutant mice. Importantly, this result indicates that Ca$_v$3.1 T-type calcium channels are critical excitatory ion channels that control the overall neural activity along with the brain state. In other words, mutant mice exhibit a less clear distinction in the neural activities associated with wakeful and unconscious states.

## Under ethanol, MD neurons lacking Ca$_v$3.1 show no burst and a wake state-like neural activity

In order to identify the mechanism linked to Ca$_v$3.1 mutant mice's ethanol-resistant phenotype, we recorded neural firing of neurons during the FWT and following a hypnotic dose of ethanol (3.0 g/kg, i.p. injection). We focused on the fLOM as it is most analogous to the classical LORR and showed the most consistency between animals (*Hwang et al., 2010*). fLOM also illustrates best the acute effect induced by ethanol before secondary metabolization enters into play.

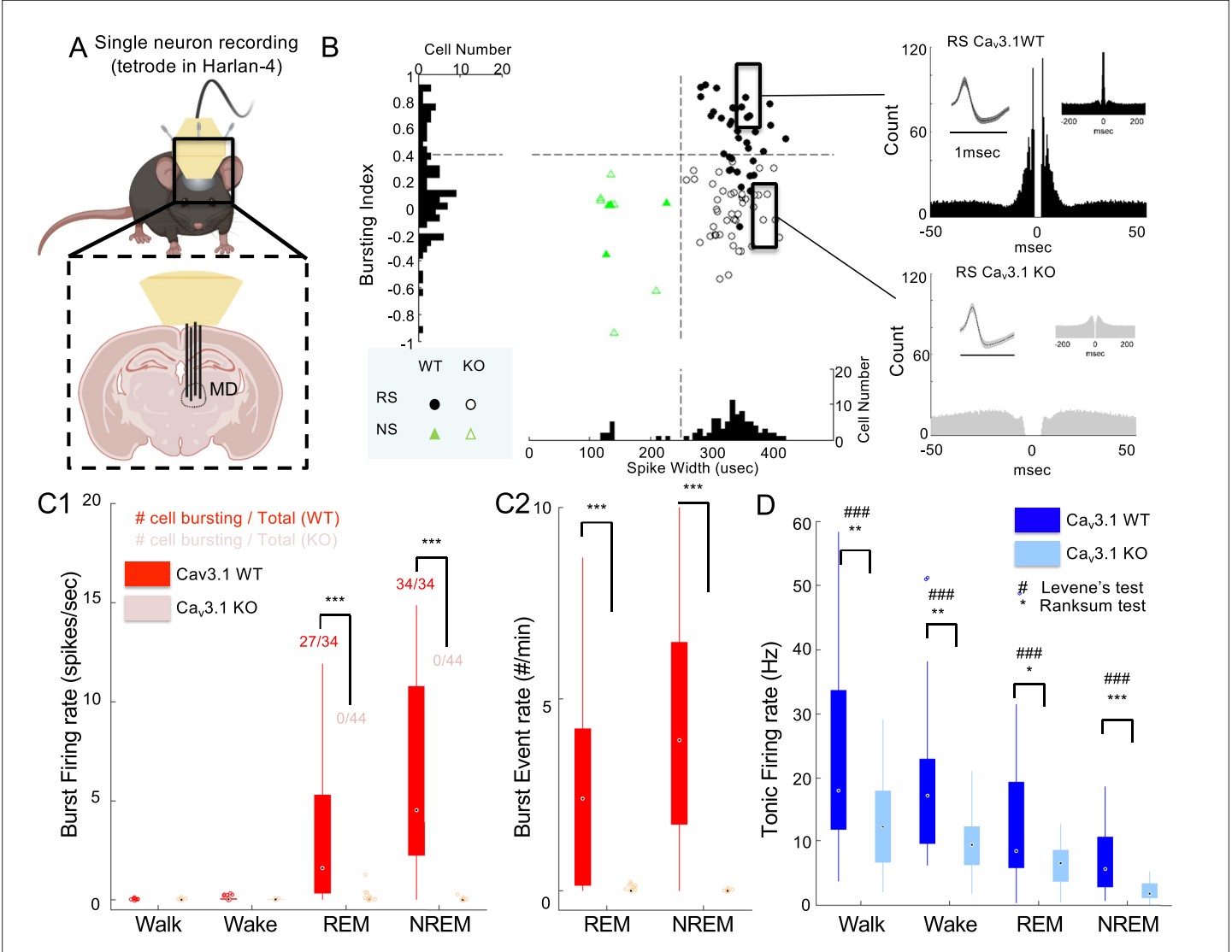

**Figure 3.** Lack of Ca$_v$3.1 removed burst firing and reduced neural activity and its variability in mediodorsal thalamic nucleus (MD) across natural conscious and unconscious states. (**A**) Mice were implanted unilaterally with four tetrode wires to record single-unit activity in the MD while in home cage (wake, sleep: non-rapid eye movement [NREM], rapid eye movement [REM]) and forced walking task (FWT) under ethanol (walk). (B - Left panel) Scatter distribution of spike width vs bursting index of MD regular spiking (RS, round shape) and narrow spiking (NS, triangle shape) neurons. Ca$_v$3.1 wild-type (WT) and Ca$_v$3.1 knockout (KO) neurons are marked as filled and empty shapes, respectively. The histogram of the pooled Ca$_v$3.1 WT and Ca$_v$3.1 KO distribution is projected on each axis. (B - Right panel) Representative autocross-correlograms of an RS neuron showing the presence and absence of fast spiking interval (burst firing) in Ca$_v$3.1 WT and mutant mice, respectively. (**C**) Boxplots of Ca$_v$3.1 wild and mutant burst firing rate (spikes/s; burst spikes-only averaged over a state duration) (**C1**) and burst event rate (#/min; number of burst events averaged over a state duration; see burst definition in Methods section) (**C2**) in RS neurons of the MD during NREM sleep, a stage known for the presence of bursting firing mode in thalamic neurons, for Ca$_v$3.1 WT and Ca$_v$3.1 KO mice. The inset numbers in C1 indicate the number of neurons showing burst firing (more than 1 event in 10 min) over the total number of single neurons identified. (**D**) Boxplots of Ca$_v$3.1 wild and mutant tonic firing rate (spikes/s) in RS neurons of the MD during walking (FWT), wake (home cage), REM and NREM sleep (home cage) for Ca$_v$3.1 WT and Ca$_v$3.1 KO mice. Group and brain state effect and interaction were assessed using a two-way repeated-measures analysis of variance (ANOVA). For post hoc, two-sample rank-sum test comparison *, **, and *** indicate p-value lower than 0.05, 0.01, and 0.001, respectively. For the two-sample Levene's test for homoscedasticity, #, ##, and ### indicate p-value lower than 0.05, 0.01, and 0.001, respectively. Pearson's rank-sum correlations between brain state and total firing for Ca$_v$3.1 WT and Ca$_v$3.1 KO are indicated above boxplots.

The online version of this article includes the following figure supplement(s) for figure 3:

**Figure supplement 1.** Representative positioning of single-unit recording in the mouse mediodorsal thalamic nucleus (MD).

**Figure supplement 2.** Representative putative neurons and burst firing properties in mediodorsal thalamic nucleus (MD).

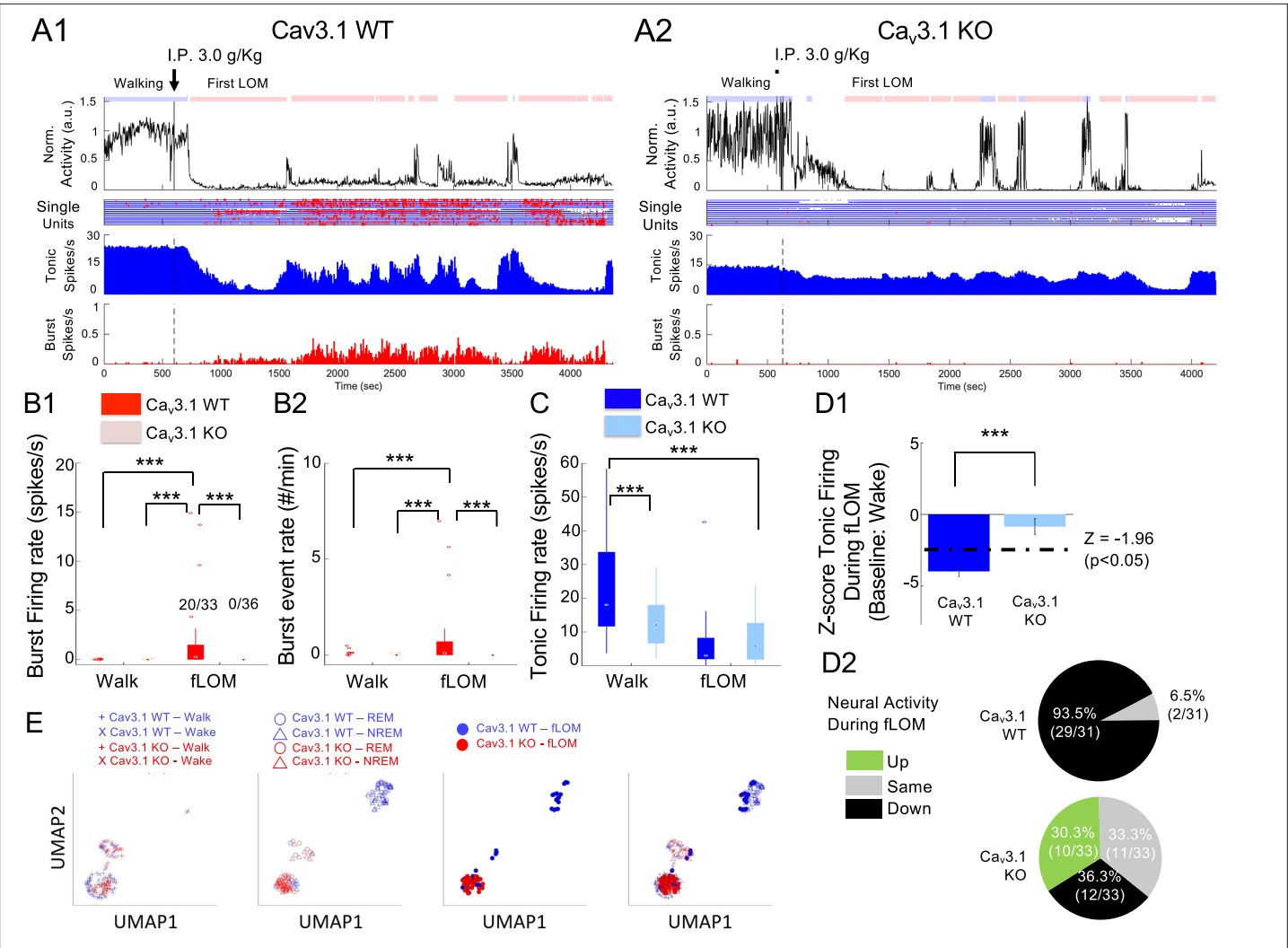

**Figure 4.** Resistance to the loss of consciousness in $Ca_v3.1$ mutant is associated with maintenance of neural activity and absence of burst. (**A**) Representative time plot for, from top to bottom, the normalized activity, single-unit raster plot (blue and red dots for tonic and burst firing, respectively), population mean firing (spikes/s), and burst-to-total spike ratio (%) for $Ca_v3.1$ wild-type (WT) (**A1**) and $Ca_v3.1$ knockout (KO) mice (**A2**). (**B**) Boxplots of $Ca_v3.1$ wild and mutant burst firing rate (spikes/s) (**B1**) and burst event rate (#/min) (**B2**) in regular spiking (RS) neurons of the mediodorsal thalamic nucleus (MD) during forced walking task (FWT) walk (pre intraperitoneal [i.p.] injection) and during FWT first loss of movement (fLOM, post i.p. injection) for $Ca_v3.1$ WT and $Ca_v3.1$ KO mice. The inset numbers in B1 indicate the ratio of the number of neurons showing burst firing over total neurons. Multiple comparisons were performed using two-sample rank-sum test or paired signed-rank test with Holm-Bonferroni correction. *** indicates a p-value<0.001. (**C**) Boxplots of tonic firing rate (spikes/s) in RS neurons of the MD during FWT walk (pre i.p. injection) and during FWT fLOM (post i.p. injection) for $Ca_v3.1$ WT and $Ca_v3.1$ KO mice. Group and brain state effect and interaction were assessed using a two-way repeated-measures analysis of variance (ANOVA). Post hoc multiple comparison performed using two-sample rank-sum test or paired signed-rank test with Holm-Bonferroni correction. *** indicates a p-value<0.001. (**D**) Normalized Z-score firing during fLOM with respect to wake state (home cage) firing (**D1**) mean and standard deviation is shown for WT (n = 31) and KO (n = 33). WT and mutant distribution and cell count based on fLOM Z-score showing increase (>1.96), no change (<1.96 and >−1.96) or decrease (<−1.96) in firing in $Ca_v3.1$ WT and $Ca_v3.1$ KO mice (**D2**). (**E**) UMAP (uniform manifold approximation and projection) two-dimensional representation of wakeful states (walk: +symbol; wake: x symbol; left panel), sleep states (rapid eye movement [REM]: empty triangle symbol; non-rapid eye movement (NREM): empty round symbol; middle panel) and fLOM state (filled round symbol; right panel) of $Ca_v3.1$ WT (blue symbols) and $Ca_v3.1$ KO (red symbols) mice. The all-state overlay is depicted on the far-right panel.

The online version of this article includes the following figure supplement(s) for figure 4:

**Figure supplement 1.** Mediodorsal thalamic nucleus (MD) neuron activity remains within wakefulness level during first loss of movement (fLOM) in $Ca_v3.1$ mutant.

Under ethanol, we observed that in WT mice, a majority of neurons showed burst firing mode (*Figure 4A1*; 20/33 bursting neurons). We found a significantly higher burst event rate (*Figure 4B1*; p<0.0001, rank-sum test with Holm-Bonferroni correction) and in the ratio of burst-to-total spike (*Figure 4B2*; p<0.0001, rank-sum test with Holm-Bonferroni correction) comparing walk (awake active) to fLOM (unconscious, unresponsive). Mutant neurons, consistent with NREM data, did not show burst firing during fLOM (0/36 bursting neurons).

Notably, in WT, we observed that ethanol induced a significant decrease in total firing from walking to fLOM states (*Figure 4A1 and C*; p<0.0001, rank-sum test with Holm-Bonferroni correction) and well below wakefulness level (home cage awake state). As in sleep, we found that a majority of RS neurons showed decreased tonic firing (total number of spikes) together with an increase in burst firing, indicating a switch in firing mode under ethanol sleep. Interestingly, the mutant mice did not show a significant decrease in total firing (*Figure 4C*; p=0.130, rank-sum test with Holm-Bonferroni correction) and showed no bursts as in sleep.

We quantified the change in activity of individual neurons using Z-score normalized to the home cage wakeful state. Here, we also observed that WT RS neurons showed a significantly reduced Z-score under ethanol fLOM, whereas in mutant mice, cells did not (*Figure 4D1*; normalized from home cage wake state; $t(62) = -5.1400$, p<0.0001, Student's t-test). Remarkably, we found that a majority of WT MD neurons (29/31) showed individual significantly decreased Z-scores (*Figure 4D2*; Z-threshold defined from a p-value of 0.05). During fLOM, mutant RS neurons subdivided into three populations (*Figure 4D2*) with decreasing (12/33, 36.4%), maintaining (11/33, 30%), and increasing (10/33, 30.3%) activity as measured by the Z-score with respect to wakefulness (i.e. home cage wake state). These results were consistent in individual mice (*Figure 4—figure supplement 1A*) and the distribution of neural population spiking (*Figure 4—figure supplement 1B*), validating that a significant drop in neural activity is associated with LOM.

Finally, we asked whether the firing modes and properties (tonic firing rate, burst firing rate; see supplementary methods) of single MD neurons would form distinct qualitative representations of 'brain stages' using a lowered dimensional uniform manifold approximation and projection (UMAP) representation (*McInnes et al., 2018*). We observed that for awake and active (i.e. walk), the brain state representation formed two adjacent clusters that confounded both wild and mutant neurons (*Figure 4E*, left panel). The REM and NREM states, the WT neurons formed two additional interconnected clusters, whereas the mutant neurons tended to overlap with the clusters attributed to the 'awake' brain state (*Figure 4E*, second to left panel). Ethanol-induced fLOM, similarly to REM and NREM clusters, was distinct from awake clusters in WT mice and overlapped with the NREM clusters (*Figure 4E*, third to left panel). Here also, mutant MD neurons showed overlap with the awake clusters rather than the 'low consciousness' brain states. These results indicate that the firing mode and properties could define a brain state representation that shows distinctions in levels of consciousness. Moreover, the mutant showed a representation of 'low consciousness' states overlapping with WT 'awake' states consistent with the hypothesis of resistance to loss of consciousness.

Altogether, these results indicate that, in WT, ethanol induced a strong reduction in neural activity and a switch to bursting firing mode correlated with loss of consciousness. However, under ethanol, MD mutant neurons maintained their activity to a level within home cage wake state without switching to bursting. This indicates that the drop in neural activity under ethanol is modulated by $Ca_v3.1$ T-type calcium channels. In its absence, MD mutant neurons display an overall reduced activity in all brain states; however, under ethanol, they remain within state wakeful levels.

## Under ethanol, 20 Hz neurostimulation of MD induces mutant-like resistance to loss of consciousness in WT mice

We observed that the maintenance of neural activity in MD excitatory neurons might be at the origin of the ethanol resistance in mutant mice. We hypothesized that artificially maintaining MD neural activity within the wakeful level would sustain consciousness under ethanol. In addition, we hypothesized that the triggering of burst firing under ethanol would potentiate loss of consciousness under ethanol. To test this possibility, we used electric and optogenetic stimulations during the FWT in WT mice under a hypnotic dose of ethanol.

MD neurons in WT mice showed a spike firing range of 0–50 Hz with an average neural firing around ~20 Hz during home cage wakefulness (*Figure 4—figure supplement 1B*). Using the 20 Hz

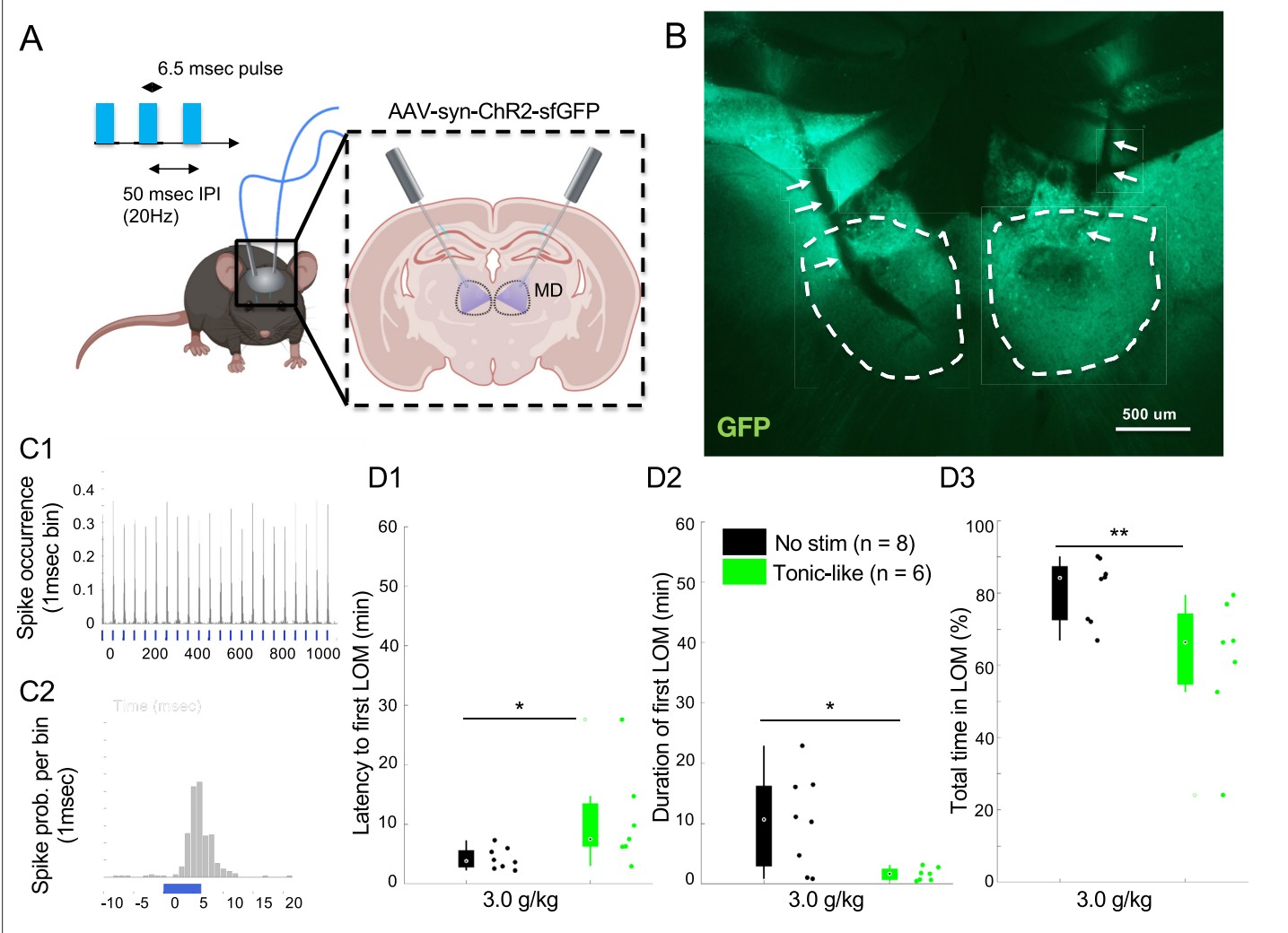

**Figure 5.** Optogenetic 20 Hz stimulation of mediodorsal thalamic nucleus (MD) in wild-type (WT) mice mimics ethanol resistance. (**A**) Mice were transduced bilaterally in the MD with an AAV-SYN-CHR2-sfGFP and implanted with bilateral optic fibers targeting MD with an entry angle of 30 degree relative to the sagittal plane. We used a stimulation protocol of tonic-like pulses at 20 Hz with 50 ms inter-pulse interval (IPI) and 6.25 ms pulse duration. (**B**) Representative expression of ChR2-sfGFP in the MD (dashed white lines) with fiber optic ending (white arrows). (**C**) In vivo response of MD regular spiking (RS) neurons to the 20 Hz tonic stimulation protocol using 6.25 ms pulse at 20 Hz and laser power of 6.0 mW (**C1**); magnification of the peristimulus response of the neuron around the laser pulse (1 ms bin; **C2**). (**D**) Latency to first loss of movement (LOM) (**D1**), duration of the first LOM (**D2**), and total time spent in LOM state (**D3**) over a recording duration of 1 hr post i.p. injection of 3.0 g/kg of ethanol are shown for the control group (no stim) and stimulated group (tonic-like). * is for p<0.05, ** is for p<0.01, *** is for p<0.001. Data is represented as boxplot with individual mice as scatter plot.

The online version of this article includes the following figure supplement(s) for figure 5:

**Figure supplement 1.** Optogenetic stimulation-induced sustained response in mediodorsal thalamic nucleus (MD).

**Figure supplement 2.** Positioning summary of optic fiber cannula used for optogenetic stimulation.

**Figure supplement 3.** Positioning summary of electrodes used for electric stimulation.

**Figure supplement 4.** Electrical 20 Hz stimulation of the mediodorsal thalamic nucleus (MD) increases ethanol resistance in wild-type (WT) mice.

**Figure supplement 5.** 1 s ON-OFF optogenetic inhibition of mediodorsal thalamic nucleus (MD) increases loss of movement (LOM) duration in wild-type (WT) mice.

stimulation (*Figure 5A*, inter-pulse interval = 50 ms, pulse width = 6.25 ms) in MD neurons transduced with excitatory channelrhodopsin (*Gangadharan et al., 2016*; *Lee et al., 2019*) (aav-SYN-ChR2-sfGFP; *Figure 5B* and *Figure 5—figure supplements 1 and 2*; see Methods), we observed an increase in ethanol resistance, which was demonstrated by a significant increase in latency to

fLOM (*Figure 5D1*; Z(13) = –2.372, p=0.013; rank-sum test). The duration of fLOM (*Figure 5D2*; Z(13) = 2.256, p=0.020; rank-sum test) and total time spent in LOM (*Figure 5D3*; Z(13) = 2.488, p=0.009; rank-sum test) were also significantly reduced. We verified that optogenetic stimulation of MD neurons at 20 Hz (*Figure 5—figure supplement 1*) induced action potentials at the same frequency with a latency response of about ~5 ms (*Figure 5—figure supplement 1B2*; pulse width = 6.25 ms). We also observed that, although marginally higher, optogenetic stimulation did not induce any significant increase in locomotor activity (*Figure 5—figure supplement 1C*; F(1,12) = 3.6232, p=0.0812) in the control and stimulated group, which indicates that the stimulation-induced increase in ethanol resistance was not due to an increase in locomotor activity.

In order to validate this observation, we then bilaterally implanted mice with twisted wires for bipolar, local electric stimulation of MD (*Figure 5—figure supplement 3*). As in the optogenetics experiment, we used a continuous pulse train of 20 Hz electric stimulation (*Figure 5—figure supplement 4A*; upper panel; inter-pulse interval = 50 ms, pulse width = 1 ms) in addition to a burst-like stimulation (*Figure 5—figure supplement 1A*; lower panel; 4 pulses at 4 ms interval and interburst interval of 1 s) that showed, respectively, tonic-like and burst-like entrainment in thalamic neurons (*Lee et al., 2012*). We observed that our 20 Hz tonic-like stimulation significantly increased the latency to fLOM (*Figure 5—figure supplement 1B1*; tonic-like vs no stim.: p=0.008; tonic-like vs burst-like: p=0.007; rank-sum test with Holm-Bonferroni correction) and significantly decreased the total time spent in LOM (*Figure 5B3*; tonic-like vs no stim.: p=0.008; tonic-like vs burst-like: p=0.007, rank-sum test with Holm-Bonferroni correction). No significant changes in the duration of the fLOM were observed (duration of fLOM) (*Figure 5B2*; tonic-like vs no stim.: p=0.917; tonic-like vs burst-like: p=0.606; rank-sum test with Holm-Bonferroni correction).

Interestingly, we observed that burst-like electrical stimulation (*Figure 5—figure supplement 4A*; lower panel; ISI = 4 ms, inter-burst interval = 1 s, pulse width = 1 ms) did not induce any significant change in ethanol sensitivity compared to the no stimulation group (latency to fLOM: p=0.365; duration of fLOM: p=0.835; total time spent in LOM: p=1; rank-sum test with Holm-Bonferroni correction). This result suggests that burst firing alone might not have a role in ethanol resistance.

Altogether, these results suggest that the maintenance of MD firing at wakefulness level (20 Hz) causally drives resistance to loss of consciousness after a hypnotic dose of ethanol. Burst-like stimulation alone did not promote or reduce loss of consciousness. This result supports the idea that neural activity maintenance in MD promotes the maintenance of consciousness even under heavy sedatives.

## Discussion

In this work, we identified that the neural activity in MD plays a causal role in the maintenance of consciousness. Whole body $Ca_v3.1$ KO and MD-specific $Ca_v3.1$ KD mice showed resistance to loss of consciousness induced by hypnotic dose of ethanol. In WT mice, MD neurons demonstrated a reduced firing rate in natural (sleep) and ethanol-induced unconscious states compared to awake states. This neural activity reduction was impaired in KO mice. In particular, transition to an unconscious state was accompanied by a switch of firing mode from tonic firing to burst firing in WT mice, whereas this mode shift disappeared in KO mice. Finally, optogenetic or electric stimulations of the MD after ethanol injection were sufficient to induce a resistance to loss of motion, supporting that the level of neural firing in the MD is critical to maintain conscious state and delay unconscious state. We showed that the expression of $Ca_v3.1$ T-type calcium channels in MD is a cellular modulator associated with this effect.

### MD is a modulator of consciousness

The role of MD in perception, attention (*Courtiol and Wilson, 2014*), and emotional control (*Lee et al., 2012*; *Paydar et al., 2014*) has been the dominant focus thus far. The recent investigations on thalamic control of consciousness revealed that nuclei within dMT hold an important modulatory function in the interaction of attention and arousal (*Schiff, 2008*; *Saalmann, 2014*). Particularly, the CM, and not VB, showed rapid shifts in LFP preceding brain state transitions such as NREM and propofol-induced anesthesia (*Baker et al., 2014*). The CL was implicated in the modulation of arousal and improvement of consciousness during seizure (*Gummadavelli et al., 2015*), and the PVN showed critical involvement in wake/sleep cycle regulation (*Colavito et al., 2015*). The mediodorsal

thalamic nucleus, however, has rarely been included as a possible pathway in the direct modulation of consciousness (*He et al., 2015*). The MD receives projections from various parts of the basal forebrain (*Schiff, 2008*) and brainstem nuclei, such as the pedunculopontine nucleus, that control the ascending pathway of arousal and attention (*Sarter and Bruno, 1999*). The MD is known to innervate the limbic region, basal ganglia, and medial prefrontal cortex (*Cassidy and Gale, 1998*), and increased activity in MD might modulate the stability of cortical UP states (e.g. awaken, aroused, and attentive states) and synchronization (*Schiff, 2008*; *Schiff et al., 2014*). Thus, MD might be a major hub involved in cortical state control and brain state stabilization.

Supporting the brain state stabilization theory and the ethanol resistance of Ca$_v$3.1 mutants, *Choi et al., 2015*, demonstrated that the loss of Ca$_v$3.1 T-type calcium channel reduced the bilateral coherence between PFC and MD under ketamine anesthesia and ethanol hypnosis, especially in the delta frequency bands. More importantly, under propofol anesthesia, *Bastos et al., 2021*, showed that ILN and MD stimulation lead to increased wake-up subscore and arousal, together with an increase in cortico-cortico and thalamocortical slow (delta) frequency power.

In the present study, we observed that MD KD (*Figure 2A*), but not VB KD (*Figure 2—figure supplement 1*), of Ca$_v$3.1 increased and is associated (*Figure 2D*) with ethanol resistance in mice. We found that MD neurons in Ca$_v$3.1 mutant mice exhibited tonic firing within the range of wakefulness (*Figures 3 and 4*), indicative of resistance to ethanol and wake-like brain state. In addition, we found a strong association between the normalized tonic firing in MD and the arousal through brain states (i.e. walk to wake to sleep states), supporting that MD tonic firing could be interpreted both as a thalamic readout and as a modulator of the brain state (*Alkire et al., 2008*; *Figure 3*). Finally, direct optogenetic and electric MD stimulation increased resistance to loss of consciousness in WT mice (*Figure 5*, *Figure 5—figure supplement 4*). To our knowledge, this is the first report demonstrating the causal involvement of the mediodorsal thalamic nucleus in the modulation of wakefulness and the resistance to ethanol-induced loss of consciousness in mice.

## Ca$_v$3.1 T-type calcium channels drive thalamic firing mode and activity

The decrease of absolute firing rate observed in thalamic neurons of Ca$_v$3.1 mutant mice supports the polyvalent role of Ca$_v$3.1 in controlling both burst and tonic firing in the thalamus. Ca$_v$3.1 channels are major contributors to excitability, and in their absence or blockade, lead to reduced neural excitability and stability and lower tonic relay of thalamocortical cells under wake-like state (*Deleuze et al., 2012*; *Tscherter et al., 2011*). The burst and tonic firing-mediated response of thalamic neurons under sensory stimulation and under the control of thalamocortical layer 6 projecting neurons was found to recruit Ca$_v$3.1 T-type calcium channels to differentiate salient novel stimuli vs complex coded information (*Mease et al., 2014*). Therefore, the nonlinear amplification and regularization of excitatory postsynaptic potentials by Ca$_v$3.1 T-type calcium channels through complexes such as with metabotropic-glutamatergic receptor 1 (*Hildebrand et al., 2009*) or the role of a 'T window' (*Crunelli et al., 2014*) would explain how the lack of T-current in mutant mice could result in an overall reduced excitation of thalamic neurons. Ca$_v$3.1 T-type is therefore a major excitatory ion channel of the central thalamic neurons.

## The lower variability in MD firing reflects ethanol resistance in Ca$_v$3.1 mutant mice

Under acute hypnotic dose of ethanol, two mechanisms might favor the reduction in firing in MD: (1) an increase in synaptic and extra synaptic GABAergic inhibition (*Jia et al., 2008*) and/or (2) reduced NMDA synaptic transmission (*White et al., 1990*). The presence of burst firing during fLOM, and during LOM in general, supports that MD neurons might have been subject to GABA receptor-mediated hyperpolarization, a necessary condition for the de-inactivation of Ca$_v$3.1 T-type burst. However, considering the dramatic difference in tonic firing observed during the FWT following i.p. injection of ethanol, the change in tonic firing in MD was the focus of our analysis.

We observed a reduction in neural firing under ethanol sleep conditions in WT mice (*Figure 4C, D1 and D2*), suggesting that low firing levels should be associated with a state of low consciousness as observed during NREM sleep. Mutant RS neurons in MD showed an overall lower excitability and variability of firing in various natural conscious and unconscious states compared to WT mice. Remarkably, Ca$_v$3.1 mutant mice exhibited a clear increase in locomotor activity and an increased resistance

to ethanol. The general lower firing rate and the high 'arousal' observed in mutant mice suggest that the relative change from state to state in tonic firing in MD, and not the absolute value of firing, might be a better correlate of change in brain state in the mice. Our optogenetic and electrical stimulation showed that sustained tonic-like stimulation in the MD at 20 Hz (*Figure 5A*), a physiologically relevant firing rate in wake state (*Figure 4*), could increase ethanol resistance in WT mice. Reducing MD firing using phasic inhibition under ethanol, potentially leading to inhibition and rebound burst (*Shao et al., 2022*), could also increase the duration of the fLOM in WT mice injected with a lower dose of ethanol (*Figure 5—figure supplement 5*; 2.0 g/kg). We propose that the relative change in firing rate in MD RS neurons might be an important driver and indicator of the change of transition in and out of consciousness, as demonstrated for other nuclei of the dMT (*Baker et al., 2014*; *Schiff, 2008*; *Gummadavelli et al., 2015*; *Colavito et al., 2015*). Therefore, the low variability in firing of MD in Ca$_v$3.1 mutant mice might be the driving force for the higher resistance to loss of motion under ethanol. In mutants, brain states might be less distinguishable, leading to frequent sleep stage switches (*Lee et al., 2004*) or resistance to unconsciousness (*Choi et al., 2015*).

## Ca$_v$3.1 T-type calcium and burst during low conscious state

Burst, as a result of Ca$_v$3.1 T-type calcium channel de-inactivation/activation, is thought to control the gating of sensory-motor stimuli (*Guido et al., 1992*; *Montemurro et al., 2007*) and modulate attention toward novel stimuli rather than the transmission of details (*Mease et al., 2014*; *Guido et al., 1992*; *Bereshpolova et al., 2011*). Previous reports highlighted the importance of burst in the stabilization of low levels of consciousness (*Kim et al., 2001*; *Anderson et al., 2005*; *Lee et al., 2004*), suggesting a direct role for burst, while no mention of the importance of tonic firing was made. We found that the propensity for burst during ethanol-induced LOM (*Figure 4A1, B1 and B2*; fLOM: 20/33 bursting neurons; 0.79±1.63 burst events/min) was lower than in NREM (NREM: 34/34 bursting neurons; 5.76±5.51 burst events/min) and higher than during wakefulness (wake: 5/36 bursting neurons; 0.16±0.31 burst events/min). In addition, burst-like electrical stimulation of MD did not significantly affect ethanol resistance (*Figure 5—figure supplement 3*). Although burst-like stimulations are highly artificial and do not recruit T-current and associated mechanisms following low-threshold burst, they allow for the reproduction of the influence of TC burst firing on target centers (*Lee et al., 2012*), including thalamocortical and thalamo-thalamic efferents.

Interestingly, under lower doses of ethanol (i.p. injection of 2.0 g/kg of ethanol), mutant and WT alike showed similar levels of resistance to ethanol. We observed that applying a phasic inhibition to MD neurons in WT under 2/0 g/Kg of ethanol, a protocol capable of inducing partial silencing (*Paz et al., 2013*) and rebound bursts (*Abdelaal et al., 2022*; *Figure 5—figure supplement 5*; 1 s ON-OFF using archaerhodopsin-mediated inhibition), did significantly increase fLOM duration mostly (Z(13) = –2.214, p=0.022, rank-sum test). This result supports that, in the context of hypnotic dose of ethanol, the apparition of bursts might correlate with unconscious state stability rather than induction. Burst stimulation without inhibition did not have this effect (*Figure 5—figure supplement 4*). Currently, our data does not allow us to formulate any clear conclusion on the direct role of burst events during fLOM. We propose that the absence of bursts and an accompanying effect of maintenance of tonic firing under ethanol in MD was responsible for the observed increase in resistance and maintenance of activity in Ca$_v$3.1 mutant mice.

## A bidirectional modulation of Ca$_v$3.1 expression and alcoholism

In humans, mutations of Ca$_v$3.1 T-type channels are associateed with mental disorders, including cerebellar ataxia, absence seizure, schizophrenia, and autism (*Lory et al., 2020*). Remarkably, mutation in voltage-gated calcium channels, including Ca$_v$3.1, leads to ethanol resistance and alcohol-seeking behavior (*Shin et al., 2005*). Reversibly, chronic exposure to ethanol intake is known to impair sleep (*Ehlers and Slawecki, 2000*) and increase ethanol resistance. Previous studies have found an alteration in T-type calcium channel expression following chronic exposure to ethanol in nonhuman primates (*Carden et al., 2006*), suggesting that a reduced T-current and the resulting sustained thalamic tonic firing could be a possible mechanism for early stage ethanol resistance in alcoholic subjects, which increases the conversion probability from casual to compulsive consumption of ethanol. The lack of burst and sustained tonic firing might impair the stabilization of sleep, and in turn, chronic sleep impairments might engage addiction-related networks. Mechanisms such as adenosine receptor

depreciation (*Clasadonte et al., 2014*; *Naassila et al., 2002*) or GABA-receptor potentiation (*Jia et al., 2007*; *Allan and Harris, 1987*; *Suryanarayanan et al., 2011*) would enhance ethanol resistance and addiction (*Koob et al., 1998*; *Schulteis and Liu, 2006*), spiraling into further sleep fragmentation, memory consolidation deficit, impulsivity, and other impairments associated with alcoholism.

## Methods

### Animals

Ca$_v$3.1 heterozygous mice (*Cacna1g$^{+/-}$*) were maintained in two genetic backgrounds, 129/svjae and C57BL/6J. All experiments used Ca$_v$3.1 homozygous mice (*Cacna1g$^{-/-}$*) KO mice, and their WT littermates in the F1 hybrid generated by mating Ca$_v$3.1 heterozygous mice (*Cacna1g$^{+/-}$*) from these two genetic backgrounds. Mice were maintained with free access to food and water under a 12 hr light/12 hr dark cycle, with the light cycle beginning at 8:00 AM. Animal care was provided, and all experiments were conducted in accordance with the ethical guidelines of the Institutional Animal Care and Use Committee of the Institute of Basic Science and the Korean Institute of Science and Technology. All experiments were conducted using 12- to 16-week-old male mice. For group comparison and grouping, mice were randomly assigned to a group and pseudo-anonymized (mouse number and treatments were traceable through a spreadsheet). Minimum group size was determined from power analysis with a β=0.8 and using pilot data and previously observed behavior in Ca$_v$3.1 KO mice. All replicates shown are biological replicates, except for the quantification of histological data where both multiple slices were used to average individual mouse quantification.

### Surgery for electrophysiological recordings and neurostimulation

The surgical implantation of electrodes (EEG, EMG, and/or tetrode Microdrive) and virus injection procedures were performed under 0.2% tribromoethanol (Avertin) anesthesia (20 mL/kg i.p.). Following anesthetic administration, mice (11-week-old for electrode implantation; 10 weeks for virus injection) were fixed in a stereotaxic device (David Kopf Instruments). For chronic recording of EEG and EMG, a stainless-steel screw electrode was fixed into the skull over the right parietal hemisphere, and an uncoated stainless-steel wire was tied to the nuchal muscle, respectively. For in vivo freely moving single-unit recording, we used a Harlan 4 Drive (Neuralynx Inc) mounted with three to four tetrode wires inserted to the caudal region of the right mediodorsal thalamic nucleus (anteroposterior, −1.4; lateral, +0.4; depth: 3.2 mm). Single tetrode wires were prepared from four twisted nichrome-formvar/PAC wires (Kanthal Precision Technology, OD 0.0127 mm) and gold-plated to achieve an impedance range of 150–400 kΩ (1 kHz, in saline solution). A period of 7 days was given to allow a complete recovery from the surgical procedure. For all chronic implantation of electrodes, an additional screw was positioned over the occipital region and used as a reference.

### Optogenetic neurostimulation

For the optogenetic experiment, 16 mice were bilaterally injected with aav9-SYN-ChR2-sfGFP virus in the mediodorsal thalamus and implanted with optic fiber guides (125 µm core diameter, Doric Lenses Inc) positioned at a 30-degree angle from the transverse plane. The mice were given 2–3 weeks to recover and to allow for the viral expression. These mice were then randomly assigned to a no stimulation (n=8) and a 20 Hz stimulation group (n=6). The mice received the stimulation immediately after being placed in the treadmill, then received the i.p. injection of 3.0 g/kg of ethanol as in other experiments. We discarded two mice due to a low viral expression found after histological analysis.

In order to measure the neural response to optogenetic stimulation, we implanted one mouse unilaterally (right MD) with a Harlan 4 Drive (four tetrodes) converging with a single optic fiber (right side, 30-degree inclination). This mouse received optogenetic stimulation in a home cage resting condition and at frequencies 1–5–10–20–40 Hz with a fixed stimulation pulse of 6.5 ms (*Figure 5—figure supplement 1*). Using these recordings, we verified the fidelity between the triggered laser stimulation and the single-unit response in the vicinity of the laser illumination. The spike per stimulation trial, spike initiation success rate, and the delayed triggered spiking (jittering) were estimated from these recordings. For all simulations, we used a high stability 473 nm (blue, MFB-III-473-AOM; Changchun New Industries Optoelectronics Technology Co., Ltd.) fiber coupled (FC) at an intensity of

6.0 mW. Laser triggering was performed using a Pulsepal pulse generator (gen1, open-source; https://open-ephys.org/pulsepal) or Master-8 (A.M.P. Instruments, Israel) pulse stimulator.

## Archaerhodopsin-mediated inhibition of MD neurons

For our phasic inhibition experiment, 16 mice were injected in MD with aav5.hSYN.eArch3.0-eYFP (University of North Carolina, Vector Core). Three mice were discarded post histological analysis due to low viral expression. This construct was favored over the halorhodopsin channel due to the long duration of the stimulation intended (60 min, 1 s pulse with a duty cycle of 50%, 0r 1 s ON-OFF sequence) and low toxicity. The mice were implanted with optic fiber guides (125 μm core diameter, Doric Lenses Inc) positioned at a 30-degree angle from the transverse plane. The mice were given 2–3 weeks to recover and to allow for the viral expression. For Arch-mediated inhibition, we used a 532 nm (Green, MGL-S-532-OEM, Changchun New Industries Optoelectronics Technology Co., Ltd) laser to deliver at ~2 mW to each fiber guide through a patch cord (SMA end-to-end; Thorlabs Inc). These mice were then randomly assigned to a no stimulation (n=7) and a 1 s ON-OFF stimulation group (n=6). The mice received the stimulation immediately after being placed in the treadmill, then received the i.p. injection of 3.0 g/kg of ethanol as in other experiments.

## Electric neurostimulation

For experiments using electrical stimulation, 18 mice were implanted with bilateral twisted dual stainless-steel wires (A-M Systems, PFA coated, 50 um diameter) targeting MD (anteroposterior, −1.4; lateral, ±0.4; depth: 3.2 mm; from bregma). The wires were minted on a custom-made 4×1 pin header connector and cemented. As in the optogenetic experiment, the mice received the stimulation immediately after being placed in the treadmill, then received the i.p. injection of 3.0 g/kg of ethanol. The mice were randomly distributed into three groups: Sham no stimulation (n=6), 20 Hz tonic stimulation (n=5; 100 μs pulse duration with IPI of 50 ms), and burst stimulation (n=7; 4× pulses of 100 μs duration at 250 Hz; inter-burst interval of 1 s). All stimulation was performed in a bipolar configuration (twisted wire, bilateral implants) and biphasic pulse (100 μA, current stimulation) using a 2100 isolated pulse stimulator (A-M Systems, Inc).

## Virus injection

WT mice (10-week-old) were placed in the stereotaxic device following 0.2% tribromoethanol anesthesia (20 mL/kg i.p.). Custom-elongated (Sutter Instrument Co.) borosilicate pipette (ID: 0.05 mm, OD: 0.07 mm, World Precision Instruments, Inc) was used to inject 0.2–0.5 μL of virus solution at a rate of 0.1 μL/min (Hamilton syringe, pump) bilaterally into the mediodorsal thalamic nuclei (anteroposterior, −1.4; lateral, ±0.4; depth: 3.2 mm). The injection pipette was then removed slowly after a diffusion period of 10 min. A period of 2–3 weeks was given to allow viral infection to settle and a complete recovery from the surgical procedure.

## $Ca_v3.1$ KD virus

For genetic KD of $Ca_v3.1$ T-type calcium channels in the MD and VB in vivo, we used a lentivirus-mediated KD injection (*Gangadharan et al., 2016*). High-titer, concentrated lentiviral vectors ($10^7$ TU/μL) expressing sh*Cacna1g* ($Ca_v3.1$ KD) (target sequence: 5′-CGGGAAGATCGTAGATAGCAAA-3′) or control shRNA (nonhuman or mouse shRNA: 5′-AATCGCATAGCGTATGCCGTT-3′) were prepared.

## Channelrhodopsin virus

Channelrhodopsin fused with superfolder GFP (ChR2-sfGFP) was designed and synthesized from published sequences using codon optimization for *Mus musculus* (DNA 2.0). To express ChR2-sfGFP in the mouse brain, the AAV vector under the control of the human Synapsin promoter (aav-SYN) was generated using PCR-amplified human Synapsin promoter. Viruses were produced with Serotype 1 or DJ (Cell Biolabs, Inc) and purified by CsCl gradients (*Feng et al., 2014*). The virus was injected at a volume of 0.5 μL in each side of the MD, followed by a bilateral implantation of optical fibers (100/125 μm, DP, Doric lens). The mice were given a period of 3 weeks to allow a strong expression of the channelrhodopsin channel following viral infection, as well as to recover from the surgical procedures.

## Ca$_v$3.1 intensity quantification

For the quantification of Ca$_v$3.1 expression in the MD, we defined ROIs centered to the left and right of the MD (2× ROIs), CL/PCN (2× ROIs), and SMT (1× ROIs; used as a control region of high Ca$_v$3.1 intensity, far from lentivirus injection). We added the CM (1× ROI central only). All ROIs were predefined using a custom script in Fiji (ImageJ, https://doi.org/10.1038/nmeth.2019) and manually rectified to match anatomical position within the nuclei. We then run a custom MATLAB script to estimate the average intensity per area for all ROIs (11 ROIs defined in total per animal for each side, left and right). All intensities were then normalized to the average intensity of the SMT (highest expression region). We then compared the normalized Ca$_v$3.1 intensity for each animal for the factors side (left, right) and KD conditions (shRNA, shCA$_v$3.1).

## Surgery for electrophysiological recordings

The surgical implantation of electrodes (EEG, EMG, and/or tetrode Microdrive) and virus injection procedures were performed under 0.2% tribromoethanol (Avertin) anesthesia (20 mL/kg i.p.). Following anesthetic administration, mice (11-week-old for electrode implantation; 10 weeks for virus injection) were fixed in a stereotaxic device (David Kopf Instruments). For chronic recording of EEG and EMG, a stainless-steel screw electrode was fixed into the skull over the right parietal hemisphere, and an uncoated stainless-steel wire was tied to the nuchal muscle, respectively. For in vivo freely moving single-unit recording, we used a Harlan 4 Drive (Neuralynx Inc) mounted with three to four tetrode wires inserted to the caudal region of the right mediodorsal thalamic nucleus (anteroposterior, −1.4; lateral, +0.4; depth: 3.2 mm). Single tetrode wires were prepared from four twisted nichrome-formvar/PAC wires (Kanthal Precision Technology, OD 0.0127 mm) and gold-plated to achieve an impedance range of 150–400 kΩ (1 kHz, in saline solution). A period of 7 days was given to allow a complete recovery from the surgical procedure.

## EEG/EMG recordings

EEG signals were amplified and band-pass filtered in the range 0.1–100 Hz. EMG signals were high-pass filtered at 70 Hz. All recordings were digitized at a sampling rate of 1 kHz (Grass Amplifiers, pClamp 9.2-Molecular devices) or at 32 kHz (Cheetah 6.5-Neuralynx) and downsampled in post-processing.

## Single-unit recording, sorting, and analysis

Electrophysiological data obtained from tetrode bundles were acquired using Digital Lynx hardware and Cheetah 6.5 (Neuralynx) at a sampling frequency of 32 kHz. Online band-pass filtering (LFP for spike sorting: 600–6000 Hz; EEG: 0.5–70 Hz; EMG: 70–4000 Hz) and spike sorting was performed using Cheetah 6.5. Off-line spike clustering and sorting were performed semi-automatically using KlustaKwik (KD Harris, http://klustakwik.sourceforge.net) and MClust 3.5 (AD Redish, http://redishlab.neuroscience.umn.edu) in MATLAB (R) (the MathWorks, Inc) or SpikeSort3D 2.5. The time stamps or spike trains associated with each identified single unit were analyzed using a customized algorithm through MATLAB (R). Single-unit characterization was performed by means of using ISI distribution, cross- and autocross-correlation histograms (e.g. bursting index, bursting mode, spectral distribution), inter- and intra-burst property analysis (e.g. intra-burst ISI, number of spikes per burst, burst spike rate) and associated spike waveform indices (e.g. peak, peak-to-valley spike width, first and second principal component). Population spiking was analyzed by means of peri-event histogram, normalized cross-correlation pairs, and phase coherency, using the Chronux toolbox (chronux.org) and custom-made codes.

The bursting index was derived as described in *Royer et al., 2012*. Namely, the burst index was estimated from the spike auto-correlogram (1 ms bin size) by subtracting the mean value between 40 and 50 ms (baseline) from the peak measured between 0 and 10 ms. Positive burst amplitudes were normalized to the peak, and negative amplitudes were normalized to the baseline to obtain indexes ranging from −1 to 1.

## Sleep monitoring and staging

Sleep scoring was based on the EEG and EMG recordings obtained from a period of 6 hr recorded in the second phase of the light cycle (12:00–18:00). We used a custom-made automatic sleep scoring system based on two previously described scoring methods for rodents (*Kohtoh et al., 2008*;

*Stephenson et al., 2009*) and organized a voting scheme for the final staging decision. All sleep scores were visually inspected and corrected by a sleep specialist.

## Immunohistochemistry

Sections of perfused mouse brain (5% formaldehyde) were intensively washed with phosphate buffer (0.1 M) and then treated with a blocking solution containing 3% normal donkey serum (Millipore) and 0.2% Triton-X (Sigma) for 40 min at room temperature. The following primary antibodies diluted in phosphate buffer were used: anti-$Ca_v$3.1 antibody (rabbit, 1:200; Alomone Labs, ACC-021, RRID:AB_2039779), anti-NeuN antibody (mouse, 1:500; Millipore, MAB377, RRID:AB_2298772), and anti-calbindin D-28k antibody (mouse, 1:3000; Swant, CB300PUR, RRID:AB_3542811). After primary antibody incubation (1 day at room temperature), sections were treated with secondary antibodies labeled with fluorescent dye (Cy3 or Cy5; 1:500, 2 hr at room temperature; Jackson). Sections with fluorescent staining were mounted in a mounting solution (VECTASHIELD with DAPI; Vector Laboratories, H-1200). Photographs were taken using either a microscope (Nikon Eclipse-Ti) or a FluoView FV1000 confocal laser scanning system (Olympus). When necessary, brightness and contrast were adjusted using the FluoView client program applied to whole images only.

## Uniform manifold approximation and projection

In order to provide a visual representation of the various brain states recorded in $Ca_v$3.1 WT and mutant mice, we combined the tonic firing rate, burst firing rate, and burst event rate into a reduced manifold representation using the UMAP method (*McInnes et al., 2018*). The version of MATLAB implementation was used with a fixed seed input.

## Drug injections

Tribromoethanol (Avertin) and ethanol were purchased from Sigma-Aldrich. All drugs were administered by i.p. injection. The surgical implantation and virus injection procedures were performed under 0.2% tribromoethanol (Avertin) anesthesia (20 mL/kg i.p.). Ethanol injections were based on a prepared stock mixture of ethanol (26%) and saline, and dosages were adjusted according to the experiments and the animal body weight (i.e. 2.0 g/kg, 3.0 g/kg, and 4.0 g/kg).

## Mouse activity classification

Mouse activity was obtained using video analysis and alternatively using an accelerometer placed on the head stage of the mouse when video wasn't available. For video analysis, after histogram filtering of the mouse's body color, the instantaneous activity was estimated as the frame-by-frame intensity difference followed by a 2D median filtering (3×3 pixel) and summed as the number of displaced pixels on camera. For the accelerometer, a zero-phase fifth-order Butterworth band-pass filter with a cutoff frequency of 0.5–20 Hz was used in order to remove the DC component; the instantaneous activity was derived as the root mean square of x-, y-, and z-axis filtered signals. The mean and standard deviation of the mean (STD) of the instantaneous activity were estimated in moving windows of 4 s duration (50% overlap).

The normalized activity index was obtained from the product of mean × STD (i.e. sustained activity and variability). Normalization was performed so that (1) complete cessation of activity approximates a value of 0 and (2) the 10 min walking baseline prior to i.p. injection averages a value of 1. A mouse was classified as not walking if its normalized activity was lower than the 95% confidence interval of baseline activity for a duration of at least 60 s and classified as walking otherwise. Non-walking states were reclassified as LOM if the mouse's normalized activity was maintained below 0.25 (lower quartile) for a duration of at least 30 s. Adjustments were performed after manual video verification.

## Statistical analysis

All statistical analyses were performed using MATLAB and SPSS 17.0 (Statistical Package for the Social Sciences). Group differences were assessed using the Student's t-test. In the case of low sample number (i.e. n<7) or distribution comparison of non-normal and/or non-equal variance number group differences were additionally confirmed using a nonparametric test (i.e. Wilcoxon rank-sum test/ signed-rank test). Multiple comparison p-value corrections were performed using a Holm-Bonferroni method. General longitudinal and group difference analysis were performed using repeated-measures

ANOVA and one/two-way ANOVA when advised. Linear correlation was performed using Pearson's correlation coefficient.

## Acknowledgements

This work was supported by the grant IBS-R001-D1 and IBS-R001-D2 from the Institute for Basic Science, Korea. We would like to thank Dr. Gireesh Gangadharan for his precious help in the editing of this manuscript.

## Additional information

### Funding

| Funder | Grant reference number | Author |
| --- | --- | --- |
| Institute for Basic Science | IBS-R001-D1 | Hee-Sup Shin |
| Institute for Basic Science | IBS-R001-D2 | Hee-Sup Shin |

The funders had no role in study design, data collection and interpretation, or the decision to submit the work for publication.

### Author contributions

Charles-francois V Latchoumane, Conceptualization, Resources, Data curation, Formal analysis, Validation, Investigation, Visualization, Methodology, Writing – original draft, Project administration, Writing – review and editing; Joon-Hyuk Lee, Conceptualization, Data curation, Formal analysis, Validation, Investigation, Methodology, Writing – review and editing; Seong-Wook Kim, Conceptualization, Formal analysis, Methodology, Writing – review and editing; Jinhyun Kim, Resources, Software, Methodology; Hee-Sup Shin, Conceptualization, Resources, Data curation, Supervision, Funding acquisition, Writing – original draft, Project administration, Writing – review and editing

### Author ORCIDs

Charles-francois V Latchoumane  https://orcid.org/0000-0001-7018-0439
Joon-Hyuk Lee  http://orcid.org/0000-0002-1984-7020
Hee-Sup Shin  https://orcid.org/0000-0003-4260-3718

### Ethics

Animal care was provided and all experiments were conducted in accordance with the ethical guidelines of the Institutional Animal Care and Use Committee of the Institute of Basic Science and the Korean Institute of Science and Technology.

Reviewer #1 (Public review): https://doi.org/10.7554/eLife.93200.3.sa1
Reviewer #2 (Public review): https://doi.org/10.7554/eLife.93200.3.sa2
Author response https://doi.org/10.7554/eLife.93200.3.sa3

## Additional files

### Supplementary files
MDAR checklist

### Data availability

Source behavioral data and representative figure and images are available on the open mendeley repository: https://doi.org/10.17632/7fr427426m.1. Additional data (heavy recording and images) can be provided upon request.

The following dataset was generated:

| Author(s) | Year | Dataset title | Dataset URL | Database and Identifier |
|---|---|---|---|---|
| Latchoumane C-FV | 2024 | Mediodorsal thalamic nucleus mediates resistance to ethanol through Cav3.1 T-type Ca2+ regulation of neural activity | https://doi.org/10.17632/7fr427426m.1 | Mendeley Data, 10.17632/7fr427426m.1 |

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
