## [Editor Report · eLife Assessment]

This **valuable** study investigates the relationship between neuronal dynamics in the thalamus and brain state modulation. The claims that a specific channel is a critical player in the regulation of brain-states and ethanol-resistance in mice are supported by **convincing** evidence. The work will be of interest to systems neuroscientists interested in brain dynamics and behavioural states.

---

## [Referee Report · Reviewer #1 (Public review)]

Summary:

This is an interesting and valuable study that uses multiple approaches to understand the role of bursting involving voltage-gated calcium channels within the mediodorsal thalamus in the sedative-hypnotic effects of alcohol. Given its unique functional roles and connectivity pattern, the finding that the mediodorsal thalamus has a fundamental role in regulating alcohol-induced transitions in consciousness state is both important for researchers investigating thalamocortical dynamics and more broadly interesting for understanding brain function. In addition, the author's examination of the role of the voltage-gated calcium channel Cav3.1 provides considerable evidence that burst-firing mediated by this channel in the thalamus is functionally important for behavioral-state transitions. While many previous studies have suggested an analogous role for these channels in sleep-state regulation, the evidence for a role of this type of bursting in sedative-induced transitions is more limited so the evidence presented is of considerable value to the field. By performing comparative experiments across multiple thalamic nuclei which have been implicated in controlling state-transitions, the authors also validate their claim and establish the unique role of the mediodorsal thalamus. Overall, this study provides substantial mechanistic insight into how the thalamus influences drug induced transitions between different states of consciousness and opens avenues for future research into how thalamocortical interactions enable brain function.

Strengths:

This study employes multiple, complementary research approaches including behavioral assays, sh-RNA based localized knockdown, single-unit recordings, and patterned optogenetic interventions to examine the role of activity in the mediodorsal thalamus in the sedative-hypnotic effects of alcohol. Experiments and analysis included in the manuscript generally appear well conceived and generally well executed. Sample sizes are sufficiently large and statistical analysis appears generally appropriate. The findings presented are novel and provide interesting insight into the role of the thalamus as well as voltage gated calcium channels within this region in controlling behavioral state-transitions induced by alcohol. In particular, the observed effects of selective knockout along with recordings in total knockout oof the voltage gated calcium channel, Cav3.1, which has previously been implicated in bursting dynamics as well as state transitions, particularly in sleep, together suggest that the transition of thalamic neurons to a bursting pattern of firing from a more constant firing is important for transition to the sedated state produced by ethanol intoxication. While previous studies have similarly implicated Cav3.1 bursting in behavioral state-transitions, the direct optogenetic interventions and single-unit recordings provide valuable new insight. These findings may also have valuable implications for the relationship between sleep process disruption associated with ethanol dependence.

Weaknesses:

While the authors have made substantial improvements to the analysis and presented important additional results, some of the methods given in the supplemental are still somewhat minimal in their description of the methods employed. In addition, the text of the manuscript still has multiple problematic issues with writing and editing that should be addressed. Such writing issues appear throughout the manuscript including in the abstract as well as in all other sections. While they do not reduce the value of the findings presented, they do make them more difficult to understand and so should be corrected.

---

## [Referee Report · Reviewer #2 (Public review)]

This study explores the role of the mediodorsal thalamus (MD) and the T-type calcium channel Cav3.1 in ethanol-induced behavioral changes, focusing on transitions between sedation and shifts in brain-states. The authors utilize genetic knockdown, optogenetic manipulation, and electrophysiological recording techniques in mice to assess the contribution of MD Cav3.1 channels to ethanol's sedative effects. The central hypothesis is that Cav3.1-mediated burst firing in the MD is essential for regulating ethanol-induced sedation and arousal transitions.

The authors' detailed responses to reviewers' comments significantly improved the manuscript, particularly regarding experimental specificity and methodological transparency. They addressed concerns about the specificity of MD knockdowns versus neighboring thalamic nuclei by adding quantifications, enhancing figure clarity, and providing lesion localization data. The revised figures, with added quantification panels, strengthened the claim that the manipulations specifically targeted the MD. Improvements in lesion validation figures and electrode placement explanations further clarified the accuracy of their methods.

One major limitation, as highlighted by Reviewer 1, is the lack of direct evidence from inhibitory optogenetic studies to validate the role of Cav3.1 channels in modulating ethanol-induced transitions in the MD. While the authors acknowledged the challenges of such experiments, citing technical issues like the inability of Cav3.1 knockout to allow rebound burst firing, the absence of these controls limits definitive causal conclusions about the MD's role. Alternative experiments with varying ethanol doses and data on tonic versus burst firing were presented, but these do not fully compensate for the missing inhibitory optogenetics, leaving some uncertainty regarding the attribution of observed behavioral effects solely to Cav3.1-mediated burst activity in the MD.

Another challenge is the complexity of distinguishing the specific contribution of the MD from that of other thalamic nuclei involved in regulating arousal and brain-states. Although additional quantification was provided to demonstrate MD specificity, control experiments targeting adjacent regions like the central lateral nucleus (CL) would have strengthened the manuscript. While the practical constraints are understandable, this limitation slightly weakens the argument regarding the MD's unique role in state transitions. The provided explanations about spatial targeting and electrophysiological methods were reasonable, but a broader set of thalamic controls would have offered a more comprehensive understanding.

Overall, the authors successfully achieved their aims, providing strong evidence that Cav3.1-mediated burst firing in the MD is crucial for ethanol-induced sedation. The knockdown experiments showed a clear reduction in ethanol sensitivity, and the behavioral assays supported the conclusion that MD Cav3.1 activity plays a key role in regulating arousal states. The combined use of Cav3.1 knockdown and optogenetic stimulation effectively linked MD activity to ethanol-induced behavioral changes. The evidence presented establishes a clear mechanistic connection between neuronal activity and behavioral responses.

The expanded discussion and clarifications in response to reviewer feedback enhanced the manuscript's coherence, and the revisions to the figures improved the transparency of the findings. Despite not implementing all the additional experiments suggested by Reviewer 1, the authors provided sufficient alternative evidence and a clear explanation of practical limitations, making their conclusions credible given the available data.

This study significantly advances our understanding of thalamic involvement in behavioral state transitions, particularly ethanol-induced sedation. By clarifying the role of Cav3.1-mediated burst firing in the MD, the research provides new insights into how specific neuronal activity patterns influence global brain states and behavioral arousal, which has implications for understanding mechanisms underlying anesthesia, sedation, and sleep regulation. Moreover, the transparency in data sharing and detailed methodological revisions make this work a valuable resource for replication or adaptation in similar studies.

---

## [Author Response]

The following is the authors’ response to the original reviews.

**Public Reviews:**

**Reviewer #1 (Public Review):**
Summary:This is an interesting and valuable study that uses multiple approaches to understand the role of bursting involving voltage-gated calcium channels within the mediodorsal thalamus in the sedative-hypnotic effects of alcohol. Given its unique functional roles and connectivity pattern, the idea that the mediodorsal thalamus may have a fundamental role in regulating alcohol-induced transitions in consciousness state would be both important for researchers investigating thalamocortical dynamics and more broadly interesting for understanding brain function. In addition, the author's examination of the role of the voltage-gated calcium channel Cav3.1 provides some evidence that burst-firing mediated by this channel in the thalamus is functionally important for behavioral-state transitions. While many previous studies have suggested an analogous role for sleep-state regulation, the evidence for an analogous role of this type of bursting in sedative-induced transitions is more limited. Despite the importance of these results, however, there is some concern that the manipulations and recording approaches employed by the authors may affect other thalamic nuclei adjacent to the MD, such as the central lateral nucleus, which has also been implicated in controlling state transitions. The evidence for a specific role of the mediodorsal thalamus is therefore somewhat incomplete, and so additional validation is needed.Strengths:This study employs multiple, complementary research approaches including behavioral assays, sh-RNAbased localized knockdown, single-unit recordings, and patterned optogenetic interventions to examine the role of activity in the mediodorsal thalamus in the sedative-hypnotic effects of alcohol. Experiments and analyses included in the manuscript generally appear well conceived and are also generally well executed. Sample sizes are sufficiently large and statistical analysis appears generally appropriate though in some cases additional quantification would be helpful. The findings presented are novel and provide some interesting insight into the role of the thalamus as well as voltage-gated calcium channels within this region in controlling behavioral state transitions induced by alcohol. In particular, the observed effects of selective knockout along with recordings in total knockout of the voltage-gated calcium channel, Cav3.1, which has previously been implicated in bursting dynamics as well as state transitions, particularly in sleep, together suggest that the transition of thalamic neurons to a bursting pattern of firing from a more constant firing is important for transition to the sedated state produced by ethanol intoxication. While previous studies have similarly implicated Cav3.1 bursting in behavioral state transitions, the direct optogenetic interventions and single-unit recordings provide valuable new insight. These findings may also have interesting implications for the relationship between sleep process disruption associated with ethanol dependence, although the authors do not appear to examine this directly or extensively discuss these implications of their findings.Weaknesses:A key claim of the study is that the mediodorsal thalamus is specifically important for the sedative-hypnotic effect of ethanol and that a transition to a bursting pattern of firing in this circuit facilitates these effects due to a loss of a more constant tonic firing pattern. Despite the generally clear observed effects across the included experiments, however, the evidence presented does not fully support that the mediodorsal thalamus, in particular, is involved. This distinction is important because some previous studies have suggested that another thalamic nucleus which is very close to the mediodorsal thalamus, the central-lateral thalamus, has previously been suggested to play a role in preventing sedative-induced transitions. Despite its proximity to the mediodorsal thalamus, the central-lateral thalamus has a substantially different pattern of connectivity so distinguishing which region is impacted is important for understanding the findings in the manuscript. While sh- RNA knockdown appears to be largely centered in the mediodorsal thalamus in the example shown, (Figure 2) this is rather minimal evidence and it is also not well explained (indeed, the relevant panels do not even appear to be referenced in the text of the manuscript) and the consistency of the knockdown targeting is not quantified. Additional evidence should be provided to validate this approach. Similarly, while an example is shown for the expression of ChR2 (Fig. 5) there seems to be some spread of expression outside of the mediodorsal thalamus even in his example raising a concern about how regionally specific this effect.The recordings targeting the mediodorsal thalamus could provide evidence of a direct association between changes in activity specifically in this part of the thalamus with the behavioral measures but there are currently some issues with making this link. One difficulty is that, although lesions are shown in Figure S5 to validate recording locations, this figure is relatively unclear and the examples appear to be taken from a different anterior/posterior location compared to the reference diagram. A larger image and improved visualization of the overall set of lesion locations that includes multiple anterior/posterior coronal sections would be helpful. Moreover, even for these example images, it is difficult to evaluate whether these are in the mediodorsal thalamus, particularly given the small size of the image shown. Ideally, an example image that is more obviously in the mediodorsal thalamus would also be included. Finally, an assessment of the relationship between the approximate locations of recorded neurons across the tetrode arrays and the behavioral measures would be very helpful in supporting the unique role of the mediodorsal thalamus. The lack of these direct links, in combination with the histological issues, reduces the insight that can be gained from this study.In addition to the key experimental issues mentioned above, there are often problems in the text of the manuscript with reasoning or at least explanation as well as numerous minor issues with editing. The most substantial such issue is the lack of clarity in discussing the mediodorsal thalamus and other adjacent thalamic nuclei, such as the central-lateral nucleus, in the author's discussion of previous findings. Given that at last one of the manuscripts cited by the authors (Saalman, Front. Sys. Neuro. 2014) has directly claimed that central-lateral, rather than the mediodorsal, thalamus is important for arousal regulation related to a conscious state, this distinction should be addressed clearly in the discussion rather than papered over by grouping multiple thalamic nuclei as being medial. As part of this discussion, it would be important to consider additional relevant literature including Bastos et al., eLife, 2021 and Redinbaugh et al., Neuron, 2020 which are quite critical but currently do not appear to be cited. Considering additional literature relevant to the function of the mediodorsal thalamus would also be beneficial. While the methods employed generally seem sound, the description in the methods section is lacking in detail and is often difficult to follow. Analysis methods such as the burst index appear to only be given a brief explanation in the text and appear not to be mentioned in the methods section. Similarly, the staining method used in Figure 2 does not appear to be described in the methods section. The most substantial case is for the UMAP approach used in Figure 4-E which does not appear to be described in the methods or even described in the main text. The lack of detailed descriptions makes it difficult to evaluate the applicability and quality of the experimental and analytical approaches. Citations justifying the use of methods such as the approach to separate regular spiking and narrow spiking neuron subtypes are also needed.Beyond the problems with content and reasoning discussed above, there are also some relatively minor issues with the clarity of writing throughout the paper for example, in the abstract the authors refer to "the ethanol resistance behavior in WT mice" but it is difficult to parse what they mean by this statement. Similarly, the next sentence "These results support that the maintenance..." while clearer, is not well phrased. Though individually minor, issues like this re-occur throughout the manuscript and sometimes make it difficult to follow so the text should be revised to correct them. There are also some problems with labels such as the labels of A1/A2 in Figure 4, which appear to be incorrect. Also, S7 has no label on the B panels. Finally, some references are not included (only a label of [ref]).
**Reviewer #2 (Public Review):**
In the current study, Latchoumane and collaborators focus on the Cav3.1 calcium channels in the mediodorsal thalamic nucleus as critical players in the regulation of brain-states and ethanol resistance in mice. By combining behavioural, electrophysiological, and genetic techniques, they report three main findings. First, KO Cav3.1 mice exhibit resistance to ethanol-induced sedation and sustained tonic firing in thalamocortical units. Second, knocked-down Cav3.1 mice reproduce the same behaviour when the mediodorsal, but not the ventrobasal, thalamic nucleus is targeted. Third, either optogenetic or electric stimulation of the mediodorsal thalamus reduces ethanol-induced sedation in control animals.Overall, the study is well designed and performed, correctly controlled for confounds, and properly analysed. Nonetheless, it is important to address some aspects of the report. The results support the conclusions of the study. These results are likely to be relevant in the field of systems neuroscience, as they increase the molecular evidence showing how the thalamus regulates brain states.
**Reviewer #1 (Recommendations For The Authors):**
Aside from the additional quantification and clarification of the analysis discussed in the weakness section, in general, the experiments included in the manuscript seem reasonable. However, I would suggest one additional experiment as well as one control, both of which are relatively straightforward optogenetic experiments, that I feel would be helpful to further improve the study. First, as the authors note, the optogenetic interventions used do not directly address the relevance of the changes in bursting patterns observed in the knockout (KO), which are by far the most robust effect, with the changes in alcohol sensitivity. One approach that could help address this would be to use patterned suppression via inhibitory opsins (e.g. halorhodopsin) to "rescue" the periods of inhibition associated with bursting in the KO. Localizing this inhibition to the mediodorsal thalamus would also lend further credence to their claim that this nuclei is the relevant circuit for their observed effects. For the control, tonic activation of the ventrobasal nucleus, as the authors did for the mediodorsal nucleus, would be beneficial to rule out the possibility that the observed effect would occur with any thalamic nucleus. In addition to these experiments, I did not note the strategy for sharing data obtained through this study so this should be added.R1 – 1: A key claim of the study is that the mediodorsal thalamus is specifically important for the sedative-hypnotic effect of ethanol and that a transition to a bursting pattern of firing in this circuit facilitates these effects due to a loss of a more constant tonic firing pattern. Despite the generally clear observed effects across the included experiments, however, the evidence presented does not fully support that the mediodorsal thalamus, in particular, is involved. This distinction is important because some previous studies have suggested that another thalamic nucleus which is very close to the mediodorsal thalamus, the central-lateral thalamus, has previously been suggested to play a role in preventing sedative-induced transitions. Despite its proximity to the mediodorsal thalamus, the central-lateral thalamus has a substantially different pattern of connectivity so distinguishing which region is impacted is important for understanding the findings in the manuscript.

R1-A1: The reviewer is right that CL has been pointed as another candidate structure with causal influence on arousal and consciousness. We have focused our efforts in including only recording single units that were from tetrode located in the MD specifically using the lesion code we explain in the method section and in response to R1 question#3. We also produced a quantification of Cav3.1 knock-down that clearly demonstrates that the KD experiment was itself specific to MD, bilaterally, and that CL to CM were minimally impacted by the knock-down process (Fig. 2C and D). Moreover, the optogenetic (fiber incidence was 30 degrees guaranteeing a central coverage rather than lateral; Fiber optic NA = 0.22) and electric stimulation (bipolar twisted electrodes, 50uA) experiments were also very selective and specific to the MD (Fig.S5). It remains clear that MD might not be the sole structure involved in the brain state control towards sedation and “anesthetic states”, and CL might be a significant contributor as well, however, we show that CL manipulations were rather irrelevant in our experiments (Fig. 2, S5, S9 and S11).

R1-2: While sh-RNA knockdown appears to be largely centered in the mediodorsal thalamus in the example shown, (Figure 2) this is rather minimal evidence and it is also not well explained (indeed, the relevant panels do not even appear to be referenced in the text of the manuscript) and the consistency of the knockdown targeting is not quantified. Additional evidence should be provided to validate this approach.

R1-A2: In order to address this important question, we have created an additional panel quantification to fig2D. We have then quantified the intensity per area of Cav3.1 expression in sub zones of 4 regions of interest: MD (left, right; 2 subzones each), Centro Medial (CM; 1 subzones in total), Centrolateral/Paraventricular nucleus (CL/PCN; left, right; 2 subzones each) and the submedial nucleus (SMT; left, right; used as a control for the intensity normalization; 1 subzones in total). This panel clearly illustrates that MD was knocked-down bilaterally (p<0.001). Moreover, CM (p<0.05) and CL (p<0.01) were also partially and unilaterally knocked down, as well. This analysis confirms that our KD had a high specificity to MD.

We added the relevant figure caption and text:

[Result section, Cav3.1 silencing in the MD, but not VB, increased ethanol resistance in mice, paragraph 3]

“We then characterized the change in Cav3.1 expression following the shControl and shCav3.1 knockdown injections in three test regions MD (left and right), CM (centromedial nucleus) and CL (centrolateral nuclei, left and right side) and a negative control region SMT (submedial thalamic nuclei, left and right side). The average intensity was obtained from two coronal brain slices for each mice used in the experiment (see Methods sections, Cav3.1 Intensity quantification). Our results show that the targeting of the knockdown was very specific to the bilateral MD (p<0.001; Fig. 2D). We noted that the CM (p<0.05) and a marginal unilateral knock-down of the CL were also observed (p<0.01). Notably, we tested the correlation between the level of knock-down in MD and the total time in LOM and observed a significant association (Fig. 2D inset; R = 0.599, p = 0.018). This result highlights that the Cav3.1 knock-down was specific to MD and with an intensity associated with ethanol-induced loss of motion.”

R1-3: One difficulty is that, although lesions are shown in Figure S5 to validate recording locations, this figure is relatively unclear and the examples appear to be taken from a different anterior/posterior location compared to the reference diagram. A larger image and improved visualization of the overall set of lesion locations that includes multiple anterior/posterior coronal sections would be helpful. Moreover, even for these example images, it is difficult to evaluate whether these are in the mediodorsal thalamus, particularly given the small size of the image shown. Ideally, an example image that is more obviously in the mediodorsal thalamus would also be included. Finally, an assessment of the relationship between the approximate locations of recorded neurons across the tetrode arrays and the behavioral measures would be very helpful in supporting the unique role of the mediodorsal thalamus.

R1-A3: Related to fig.S5, we re-distributed the position of the recordings from the tetrode electrode burned positions over 3 representative coronal planes that best represent the implant positions. We also provided additional snapshots of tetrode location. To identify the positions of four tetrodes in each animal, we encoded the positions with different electrical lesion strategies as follows: 1 lesion(tetrode 1), 2 lesions while we redrew the tetrode with 100 um interval (tetrode 2), 3 lesions with 200um interval (tetrode 3), 4 lesions with 50um intervals (tetrode4). Tetrodes that were found outside of the MD delimited region were discarded post analysis. A straight relationship between the closeness of the electrode is unfortunately not possible for tetrode recording, a straight silicone probe which maintains the spatial spacing in recording would have been a better approach in that case, but unfortunately, it was not performed in our study.

R1-4: In addition to the key experimental issues mentioned above, there are often problems in the text of the manuscript with reasoning or at least explanation as well as numerous minor issues with editing. The most substantial such issue is the lack of clarity in discussing the mediodorsal thalamus and other adjacent thalamic nuclei, such as the central-lateral nucleus, in the author's discussion of previous findings. Given that at last one of the manuscripts cited by the authors (Saalman, Front. Sys. Neuro. 2014) has directly claimed that central-lateral, rather than the mediodorsal, thalamus is important for arousal regulation related to a conscious state, this distinction should be addressed clearly in the discussion rather than papered over by grouping multiple thalamic nuclei as being medial. As part of this discussion, it would be important to consider additional relevant literature including Bastos et al., eLife, 2021 and Redinbaugh et al., Neuron, 2020 which are quite critical but currently do not appear to be cited. Considering additional literature relevant to the function of the mediodorsal thalamus would also be beneficial.

R1-A4: We thank the reviewer for his comments and suggestions. We agree that the added references mentioned by the reviewers are highly relevant and should be integrated in the manuscript. We have integrated the above-mentioned references and further developed on the discussion on the role of MD relative to other thalamic nuclei (ILN and CL in particular). We believe that this better-referenced and clarified text does improve the manuscript greatly.

[introduction section, paragraph 3]

“The centrolateral (CL) thalamic nucleus has been implicated in the modulation of arousal, behavior arrest 31, and improvement of level of consciousness during seizures 32. Notably, the direct electrical stimulation of the intralaminar nuclei (ILN) and, in particular CL, promoted hallmarks of arousal and awakening in primate under propofol and ketamine propofol anesthesia.”

[Discussion section, paragraph 1]

“In this work, we identified that the neural activity in MD plays a causal role in the maintenance of consciousness. Whole body Cav3.1 KO and MD-specific Cav3.1 KD mice showed resistance to loss of consciousness induced by hypnotic dose of ethanol. In WT mice, MD neurons demonstrated a reduced firing rate in natural (sleep) and ethanol-induced unconscious states compared to awake states. This neural activity reduction was impaired in KO mice. In particular, transition to an unconscious state was accompanied with a switch of firing mode from tonic firing to burst firing in WT mice whereas this modeshift disappeared in KO mice. Finally, optogenetic or electric stimulations of the MD after ethanol injection were sufficient to induce a resistance to loss of motion, supporting that the level of neural firing in the MD is critical to maintain conscious state and delay unconscious state. We showed that the expression of Cav3.1 t-type calcium channels in MD is a cellular modulator associated with this effect.”

[Discussion section, MD is a modulator of consciousness, paragraph 2 and 3]

“The MD is known to innervate limbic region, basal ganglia and medial prefrontal cortex 50 and increased activity in MD might modulate the stability of cortical UP states (e.g. awaken, aroused and attentive states) and synchronization 9,26. Thus, MD might be a major hub involved in cortical state control and brain state stabilization.

Supporting the brain state stabilization theory and the ethanol resistance of Cav3.1 mutants, Choi et al.34 demonstrated that the loss of Cav3.1 T-type calcium channel reduced the bilateral coherence between PFC and MD under ketamine anesthesia and ethanol hypnosis, especially in the delta frequency bands. More importantly, under propofol anesthesia, Bastos et al.35 showed that intralaminar nucleus and MD stimulation lead to increased wake-up subscore and arousal, together with an increased in cortico-cortico and thalamo-cortical slow (delta) frequency power.

In the present study, we observed that MD KD (Fig. 2A), but not VB KD (Fig. S3) of Cav3.1 increased and is associated (Fig. 2D) with ethanol resistance in mice. We found that MD neurons in Cav3.1 mutant mice exhibited tonic firing within range of wakefulness (Fig. 3 and 4), indicative of resistance to ethanol and wake-like brain state. In addition, we found a strong association between the normalized tonic firing in MD and the arousal through brain states (i.e. walk to wake to sleep states), supporting that MD tonic firing could be interpreted both as a thalamic readout and a modulator of the brain state 11 (Fig. 3). Finally, direct optogenetic and electric MD stimulation increased resistance to loss of consciousness in WT mice (Fig.5 and Fig. S10). To our knowledge, this is the first report demonstrating the causal involvement of mediodorsal thalamic nucleus in the modulation of wakefulness and the resistance to ethanol-induced loss of consciousness in mice.”

R1-5: While the methods employed generally seem sound, the description in the methods section is lacking in detail and is often difficult to follow. Analysis methods such as the burst index appear to only be given a brief explanation in the text and appear not to be mentioned in the methods section.

R1-A5: We have added a clear definition in the supplementary method following the original work used:

[Supplementary Method section, Single Unit recording, sorting and analysis, last paragraph]

“The bursting index was derived as described in (Royer et al. 2012). Namely, the burst index was estimated from the spike auto-correlogram (1-ms bin size) by subtracting the mean value between 40 and 50 ms (baseline) from the peak measured between 0 and 10 ms. Positive burst amplitudes were normalized to the peak and negative amplitudes were normalized to the baseline to obtain indexes ranging from −1 to 1.” We also edited its mention in the text for clarity:

[Result section, Lack of Ca3.1 in MD neurons removes thalamic burst in NREM sleep, paragraph 2]

“[…] and a clear reduction in total bursting represented as bursting index (Fig. 3-B; ratio of spikes count <10 ms and >50 ms based on auto-cross-correlogram).”

R1-6: Similarly, the staining method used in Figure 2 does not appear to be described in the methods section.

R1-A6: The staining method can be found in the supplementary method of the paper. [supplementary method, Immunohistochemistry]

R1-7: The most substantial case is for the UMAP approach used in Figure 4-E which does not appear to be described in the methods or even described in the main text.

R1-A7: Regarding the method, the UMAP approach is described in the supplementary method document [Uniform Manifold Approximation and Projection (UMAP)]. We believe that only a succinct description was needed here considering the extent of the analysis. Regarding the inserts in the main text, we agree that the main text was lacking the description of these results and we have amended the main text to reflect a clear description of this result and what it entails. The following paragraph was added:

[Result section, Under ethanol, MD neurons lacking Cav3.1 show no burst and a wake state-like neural activity, second to last paragraph]

“Finally, we asked whether the firing modes and properties (tonic firing rate, burst firing rate; see supplementary methods) of single MD neurons would form distinct qualitative representation of “brain stages” using a lowered dimensional UMAP representation (Uniform Manifold Approximation and Projection42). We observed that for awake and active (i.e. walk), the brain state representation formed two adjacent clusters that confounded both wild and mutant neurons (Fig. 4E, left panel). The REM and NREM states, the wild type neurons formed 2 additional interconnected clusters, whereas the mutant neurons tend to overlap with the clusters attributed to the “awake” brain state (Fig. 4E, second to left panel). Ethanol induced fLOM, similarly to REM and NREM clusters, was distinct from awake clusters in wild type mice and overlapped with the NREM clusters (Fig. 4E, third to left panel). Here also, mutant MD neurons showed overlap with the awake clusters rather than the “low consciousness” brain states. These results indicate that the firing mode and properties could define a brain state representation that shows distinctions in levels of consciousness. Moreover, the mutant showed a representation of “low consciousness” states overlapping with wild type “awake” states consistent with the hypothesis of resistance to loss of consciousness.”

R1-8: Citations justifying the use of methods such as the approach to separate regular spiking and narrow spiking neuron subtypes are also needed.

R1-A8: We have added two references related to the observation of the two subpopulations of spiking neurons [Schiff and Reyes, 2012; Destexhe, 2008].

R1-9: Beyond the problems with content and reasoning discussed above, there are also some relatively minor issues with the clarity of writing throughout the paper for example, in the abstract the authors refer to "the ethanol resistance behavior in WT mice" but it is difficult to parse what they mean by this statement.

R1-A9: We addressed this issue by editing and revising the manuscript for clarity and flow.

R1-10: Similarly, the next sentence "These results support the maintenance..." while clearer, is not well phrased. Though individually minor, issues like this re-occur throughout the manuscript and sometimes make it difficult to follow so the text should be revised to correct them.

R1-A10: We thank the reviewer for highlighting this point. We have edited the overall text to improve clarity and flow.

[abstract]

These results suggest that maintaining MD neural firing at a wakeful level is sufficient to induce resistance to ethanol-induced hypnosis in WT mice.

R1-11: There are also some problems with labels such as the labels of A1/A2 in Figure 4, which appear to be incorrect.

R1-A11: We noted this issue and have rectified the figure for clarity.

R1-12: Also, S7 has no label on the B panels.

R1-A12: We thank the reviewer for pointing out this lack. We have added the y-label on the panel for clarity.

R1-13: Finally, some references are not included (only a label of [ref]).

R1-A13: We have completed the missing reference and thank the reviewer for pointing that out.

Additional commentsR1-14: Aside from the additional quantification and clarification of the analysis discussed in the weakness section, in general, the experiments included in the manuscript seem reasonable. However, I would suggest one additional experiment as well as one control, both of which are relatively straightforward optogenetic experiments, that I feel would be helpful to further improve the study. First, as the authors note, the optogenetic interventions used do not directly address the relevance of the changes in bursting patterns observed in the knockout (KO), which are by far the most robust effect, with the changes in alcohol sensitivity. One approach that could help address this would be to use patterned suppression via inhibitory opsins (e.g. halorhodopsin) to "rescue" the periods of inhibition associated with bursting in the KO.

R1-A14: Here the reviewer proposes an interesting experiment which we have attempted to perform, however, poses several technical challenges. First, the KO do not have burst firing as they are depleted from Cav3.1 low-threshold calcium channel. Therefore, under ethanol, even if there might exist a rhythmic inhibition that activates Cav3.1 channels and causes a rebound burst, the KO are unable to have it. Therefore, an optogenetic inhibition would only accentuate the total inhibition and could potentially induce an overall decrease in MD firing, resulting in an increase in LOM features. Alternatively, we showed that in a WT with low ethanol dose (where LOM induction is harder), the increased rhythmic inhibition does indeed increase significantly LOM duration and marginally decreases latency to LOM (Fig. S12), indicating that increased inhibition could indeed explain the hypothesis: “ the stronger the decrease in MD firing, the faster and longer the LOM.” The only caveat of using WT here is that optogenetic inhibition might also include rebound burst post-inhibition. Injecting bursts only did not alter the response to ethanol (Fig. S10). These results point to the role of loss of firing in MD as a main factor for LOM, and potentially the contribution of burst necessitating a concurrent inhibition/loss of firing.

We agree that inhibition in KO would further validate this hypothesis, controlling for the role of burst. We regret that we are not in the capacity to perform additional experiments involving the KO mice.

R1-15: For the control, tonic activation of the ventrobasal nucleus, as the authors did for the mediodorsal nucleus, would be beneficial to rule out the possibility that the observed effect would occur with any thalamic nucleus.

R1-A15: We agree with the reviewer that we could have added an additional region control to the gain/loss of function experiments. We would even go further as to suggest that a better control nucleus would be a high order nucleus such as PO or an unrelated sensory relay nucleus such as LGN. VB being a motor relay nucleus, could also mediate movement initiation, which could be hard to interpret. Since the complete control study for all thalamic nuclei Cav3.1 KD is outside the scope of this study, we opted not to redo these experiments and keep the focus of the manuscript on the manipulation of MD activity rather than the various available thalamic nuclei. We also do not claim that MD is the sole center able to initiate a switch in the loss of consciousness, and a more in-depth study on that matter would be clearly needed.

R1-16: In addition to these experiments, I did not note the strategy for sharing data obtained through this study so this should be added.

R1-A16: We have uploaded data and code for most figures at the following repository and provided a clearer statement regarding data sharing. We thank the reviewer for pointing out this missing element.

The link for the repository is the following:

It contains:

- Excel spreadsheet file of all behavior values, including the newly quantified Cv3.1 expression in MD/CL/SMT

- Excel spreadsheet follow-up of all MD cells (single unit; tetrode) analyzed

- Folders for all groups studied with representative figures showing EEG power over time and normalized activity (WT vs KO for 2, 3 and 4 g/kg; MDshKD vs shCTR, VBshKD vs shCTR; CHR2 NOSTIM vs STIM; ESTIM Groups and ARCH NOSTIM vs STIM)

- A1G LORRvsLOM and OPEN FIELD Matlab data

- Matlab and ImageJ Codes: single unit analysis, characterization, brain state characterization, sleep stages, LOM, open field analysis and statistical analysis.

We have added the data sharing subsection in the acknowledgements:

“Part of the analyzed data and codes are available on the open access platform, mendeley:

Latchoumane, Charles-francois (2024), “Mediodorsal thalamic nucleus mediates resistance to ethanol through Cav3.1 T-type Ca2+ regulation of neural activity”, Mendeley Data, V1, doi: 10.17632/7fr427426m.1

Additional data (large size recording and images) can be provided upon reasonable requests.”

**Reviewer #2 (Recommendations For The Authors):**
R2-1. Consciousness is a contentious subject. Even in humans, there is still intense research on the topic, not to mention animals, about which we still know very little. Moreover, consciousness is not quantified in this study, as there is no standard metric to do so. Accordingly, talking about 'modulation', 'transition', ´level ', or 'reduction' of consciousness can be misleading. Hence, it is probably safer to strictly refer to brain-states and/or stages of the sleep-wake cycle in this study and reframe it entirely around these concepts.

R2-A1. The reviewer points to an important point and we appreciate this highlight. Agreeing that the definition of consciousness is rather loose and arguably difficult to pinpoint. Here, we settle on a definition that relies on the loss of motion and loss of righting reflex. This definition is widely accepted as the “verified” state in which the absence of responsiveness (to continuous stimuli, inducing reflex or discomfort) is observed and uninterrupted by jerks and spurious movements. Additional metrics needed would be the recording of EMG to quantify atonia and EEG to the settling of a dominantly slow-wave frequency (~4 Hz; ethanol-induced sedation at theta rhythm), as shown in Fig S1A. The driver of this 4Hz frequency and its correlation has been investigated previously (e.g. Choi et al, PNAS, 2012), leading to the accepted link between LOM/LORR and loss of consciousness. Our data present the advantage of showing single neuron recordings and that LOM is a state where the lowest firing activity is present (Fig S7AB) and comparable to deep sleep state activity (Fig3D). The first LOM is the most important as it highlights the deepest loss of consciousness before the ethanol starts to be metabolized and cleared, which would be consistent between animals.

As a result, we have edited the manuscript to clarify all mentions related to brain states and states of unconsciousness.

R2-2. It is not clear why the authors focus on the mediodorsal nucleus. This should be better explained in the introduction and developed in the discussion.

R2-A2. This comment converges with the Reviewer 1 comments and we are addressing this lack in the discussion as suggested. We have addressed it with this previous comment and believe it is now clearer.

R2-3. The discussion mentions that 'increased activity in MD might modulate the stability of cortical UP state and synchronization' (pg 21). This point should be either further developed and put into context, or removed. In its current state, it does not seem to contribute much to the discussion of results.

R2-A3. We understand that the working “UP state” might not be clear enough. We have modified this sentences as follows to clarify that UP state could be either a state of where the animal is awake, aroused or attentive:

[Discussion section, MD is a modulator of consciousness, first paragraph]

“The MD is known to innervate limbic region, basal ganglia and medial prefrontal cortex 50 and increased activity in MD might modulate the stability of cortical UP states (e.g. awaken, aroused and attentive states) and synchronization 9,26. Thus, MD might be a major hub involved in cortical state control and brain state stabilization.“

R2-4. The discussion states that 'mutant mice did not exhibit a decreased arousal level (i.e. increased locomotor activity)' (pg 23). This is confusing as decreased arousal should be reflected in decreased locomotor activity.

R2-A4. We understand that the formulation of this sentence may be confusing and we have edited this portion of the text to improve quality in the revised version of the manuscript. To clarify, mutant mice do not exhibit reduced or increased arousal (not quantified, just observational), they do have a phenotypic hyperlocomotion. This comes in contrast with a lower basal firing rate in the MD, which in our interpretation, is not synonymous with lower arousal. We believe that the relative change in MD determines the change in arousal, and that the absolute firing is not indicative of arousal in itself, only in comparison.

[Discussion section, The lower variability in MD Firing reflects Ethanol Resistance in Cav3.1 mutant mice, paragraph 2]

“Mutant RS neurons in MD showed an overall lower excitability and variability of firing in various natural conscious and unconscious states compared to wild type mice. Remarkably, Cav3.1 mutant mice exhibited a clear increased locomotor activity and an increased resistance to ethanol. The general lower firing rate and the high “arousal” observed in mutant mice suggests that the relative change from state to state in tonic firing in MD, and not the absolute value of firing, might be a better correlate of change in brain state in the mice.”

R2-5. The methods (pg 27) state that two genetic backgrounds (129/svjae and C57BL/6J) were used in the study. Authors should show whether there were significant differences between those backgrounds in the key parameters assessed in the study (particularly resistance to ethanol sedation).

R2-A5. As mentioned in the method section, we only used the F1-background mice, which are the firstgeneration offspring produced by crossing 129/svjae and C57BL/6J strains. To produce F1 KO mice, we kept the heterozygote mice in two strains. We unfortunately did not study the particular difference of the respective KO of these two backgrounds; however, the pure C57BL/6J KO has been used in other studies by our group (Kim et al 2001; Na et al, 2008; Park et al., 2010). The F1 background allows us to work with mice that are less aggressive and can be handled with less inherent stress.

R2-6. It would be convenient to produce a supplementary figure associated with Figure 1C to show the same data with averages per mouse. That is, 9 points for control and 9 points for KO mice. This also applies to all cases where data is not presented per mouse but pooled between animals.

R2-A6. We have added a panel C in Figure S1, to show the scatter values for all the mice corresponding to the figure 1C. We have also generalized this presentation for all behavior graphics showing all the animals in the scatter plot next to the boxplot. We believe that this presentation increases further the transparency of the manuscript. We have then added the scatter plot for all mice in figure Fig1, Fig2, Fig5, Fig.S2, Fig.S3, Fig.S10 and Fig.S12.

R2-7. It would be informative to make a supplementary figure associated with Figure 1D to compare baseline raw activity levels (i.e., baseline walking recording) between control and KO mice. That is, do KO and control mice cover comparable distances and at similar speeds during baseline conditions? Figure 1D and Figure 4A suggest that the variability of locomotor activity is larger in KO mice. Hence, this parameter should be quantified and reported.

R2-A7. We thank the reviewer for this comment. We strived to answer to this question in the manuscript in two ways:

- We first measure the overall hyperlocomotion of the mice using the open field total distance parkoured in our mice cohorts (FigS4C). We did observe that the KO mutant showed hyperlocomotion, but not MD or VB knock-down mice. Which indicates that the hyperlocomotion component is not specific to the two thalamic nuclei studied.

- Using the forced walking task, we impose on the animal to keep a steady pace of roughly 6cm/s. This assay allows to normalize the general walking behavior to a relatively fixed pace making it comparable for all animals.

The reviewer suggested reporting the mean and variance in walking of WT and KO during baseline (prior to the ethanol I.P. injection). We believe that the two points mentioned above are sufficient to describe in a more quantitative way the WT vs KO locomotion differences. Moreover, by construction the normalized locomotion on the forced walking task will return similar means for the baseline, the standard deviation would, however, potentially show differences but would remain inconclusive.

R2-8. The legend in Figure 1 states that 'the loss of consciousness is evaluated using normalized moving index using either video analysis (differential pixel motion), on- head accelerometer-based motion, or neck electromyograms'. Authors should clarify whether these methods are equivalent and support it with data.

R2-A8. We understand the reviewer point and we have made a few modifications to the method description aligning better with what was done. For most mice, video analysis was used to obtain the moving index. When video recording was not available (2 mice), we had an accelerometer attached to the animal’s head stage which helped us derive a moving index that was similar to the video moving index. The neck electromyogram was rather used for animals implanted with the tetrodes to identify sleep stages based on local field potential frequency and muscle tone. We have then clarified the method for this matter and Figure 1 to avoid this confusion. Since no concurrent recording of both video and accelerometer was performed, we do not have the data to compute the correlation between the two measures, however, no noticeable deviation from loss of motion was observed between the two methods. We realize that this may be a weak argument, however, our observations showed that video and accelerometers returned very similar timings for loss of motion (only a few comparative instances insufficient to present a statistical comparison).

R2-9. How were spike bursts defined? The authors should try different criteria and verify the consistency of results.

R2-A9 For in vivo single unit recording, we opted for a definition that is validated from our works and others as a silencing of at least 100 ms followed by a minimum of 3 spikes with:

- First spike pairs interspike interval less than 4 ms

- Remaining spike pairs interspike interval less than 20 ms

We have performed this analysis using a minimum of 2 spikes, and varied silencing periods between 50 and 100ms, without observing significant deviation of the results. As shown in Figure S6B, with this approach we observed that the burst distribution had a majority with <10 spikes per burst. Figure S6C indicated that with a clear distribution of ISI for first spike within 2-4ms as observed in previous works (Desai and Varela, 2021; Alitto et al, 2019), importantly, not clearly capped at 4 ms, showing that the range for the first ISI might indeed be lower than 4ms for thalamic burst. Within burst spike waveforms can become very variable and the choice of 3 over 2 spikes minimum per burst stems from the aim to reduce false positive detection of ultra-short bursts, which in single unit recording remains controversial (Gray et al. 1995).

Minor:R2-10: Figure 4A2 'Cav3.1(+/+)' should presumably be Cav3.1(-/-).

R2-A10: this is correct and we have corrected the figure label [This sentence is ambiguous. What is ‘this’ that is correct?]

R2-11: Figure S2C legend states 'Post-hoc group comparison was performed using.' The sentence seems to be incomplete.

R2-A11: We have completed the sentence for clarity.

R2-12: In the methods (pg 29) virus concentration is reported as '107 TU/ul', which probably refers to 10e7.

R2-A12: We have corrected it by superscripting the power 7.

R2-13: Verify Fig 1C1 and correct Y-axis overlap between title and units.

R2-A13: We edited the figure for clarity, thank you.

R2-14: On page 24 there is a '[ref]' that probably stands for (a missing) reference.

R2-A14: the missing reference has been added.